# YOLO-based intelligent recognition system for hidden dangers at construction sites

**Hang Li[1], Peijian Jin** **[1,2]\*, Long Zhan[3], Weilong Yan[1], Shihao Guo[1], Shimei Sun[1,2]**

**1** School of Emergency Science and Engineering, Jilin Jianzhu University, Changchun, Jilin, China, **2** Jilin Provincial Key Laboratory of Fire Risk Prevention and Emergency Rescue for Building, Changchun, Jilin, China, **3** Baoye (Changchun) Construction Development Co., Ltd., Changchun, Jilin, China

\* jinpeijian@jlju.edu.cn

## Abstract

To reduce the accident rate in the construction industry, an improved YOLOv5n-based hazard recognition system for construction sites is proposed. By incorporating optimization mechanisms such as the ECA attention module, ghost module, SIoU loss, and EIoU–NMS into YOLOv5n, the system achieves both lightweight acceleration and improved accuracy. Two ultrasmall models (approximately 2.5 MBs each) were trained on a self-built dataset to detect "unsafe human behaviors" and "unsafe object conditions," achieving mAP@0.5 scores of 93.6% and 99.5%, respectively. After deployment on the Jetson Nano B01 edge platform, the system was constructed, and its high efficiency in onsite hazard detection was validated.

## Introduction

The construction industry is known for its labor-intensive nature, integration of multiple processes, complex working environments, and frequent unexpected incidents. Ensuring construction safety is crucial to the smooth operation of the industry. However, under the current traditional safety management framework, issues such as missing personnel data, unclear equipment layouts, and a lack of environmental parameters persist, resulting in new safety hazards. These hazards refer to potential risks at construction sites that are not easily observed or detected through routine inspections. They include but are not limited to nonusage of safety helmets, smoking in no-fire zones, improper equipment storage, unsafe scaffolding, and damaged cables. Therefore, developing a method for identifying construction safety hazards with high accuracy, timeliness, and stability is a key step in improving the effectiveness of construction safety accident prevention. To address these issues, object detection technology based on deep learning is an effective solution.

In recent years, object recognition using deep learning has emerged as a major focus in computer vision research. Unlike traditional techniques that rely on manual feature design and extraction, deep learning models autonomously learn features,

**Data availability statement:** All files are available on Figshare under DOI: https://doi.org/10.6084/m9.figshare.28644194.v1.

**Funding:** Jilin Provincial Department of Science and Technology, Grant number: YDZJ202101ZYTS146

leading to a significant increase in accuracy [1,2]. These deep learning detection algorithms are generally categorized into two types: region-based two-stage detection algorithms and end-to-end single-stage detection algorithms [3]. Two-stage algorithms, such as R-CNN [4], introduced by R. Girshick et al., and its more advanced variants, Fast R-CNN [5] and Faster R-CNN [6], are known for their high detection accuracy but longer processing times, making them less ideal for real-time applications. In contrast, single-stage detection algorithms such as SSD [7] and the YOLO series [8–11] are both fast and highly accurate. The YOLO series, in particular, stands out because of its simpler network architecture, superior generalization capabilities, and overall better performance, which contributes to its widespread use across various fields.

Zhang et al. proposed a lightweight detection algorithm for safety helmets and reflective vests on the basis of YOLOv5s. They introduced the ghost module to reduce model complexity, added the coordinate attention (CA) module and used the C3CBAM attention module to improve detection accuracy [12]. Cui et al. developed a helmet-wearing detection model named CA-YOLO based on YOLOv5s, incorporating the CA attention mechanism into the backbone network and proposing the PSM-EMA optimization strategy to reduce the false detection rate [13]. Alateeq et al. proposed a PPE and heavy equipment detection model based on the YOLOv5s algorithm, integrating weather conditions into the model to analyze surrounding hazards under specific weather scenarios [14]. Liang et al. enhanced YOLOv5 performance by optimizing the loss function and improving the Mosaic-9 data augmentation algorithm, leading to a helmet-wearing detection model based on YOLOv5 [15]. Yao et al. proposed an improved YOLOv5 helmet detection model by first adding a small-object detection layer, then incorporating a multispectral channel attention (MSCA) mechanism, and finally optimizing the activation function to enhance performance [16]. Li et al. introduced a PPE detection model based on YOLOv5 that incorporates a dedicated small-object detection feature layer and a DilateFormer attention mechanism to balance detection accuracy and computational efficiency [17]. Liu et al. constructed a new PPE dataset and used an SEIoU loss function combined with the CA mechanism to improve the YOLOv5 algorithm, ultimately proposing a PPE detection model [18]. Ren et al. presented a lightweight helmet detection algorithm based on an improved YOLOv5, which uses distribution shifting convolution (DSConv) layers and ghost modules to reduce the model weight, whereas SE attention was used to increase accuracy [19]. Zhang et al. integrated CSPNet and GhostNet to develop a more efficient feature extraction module, the CGMS. They further lightened the model using ghost convolution, enhanced the performance with EIoU loss and SE attention, and finally proposed a lightweight helmet detection model [20]. Wei et al. proposed a model named BiFEL-YOLOv5s by combining the BiFPN and focal EIoU loss to improve YOLOv5s performance, leading to a trained model for helmet-wearing detection [21]. Pei et al. developed a small-object detection model called SGD-YOLOv5, which integrates a deep spatial convolution module and a global attention mechanism to improve performance. They also replaced the YOLOv5 head with a decoupled head for better classification and faster convergence. The model

was evaluated on the expanded Safety Helmet Wearing Dataset (SHWD) and Smoking Behavior Detection Dataset (SBDD) [22]. To enhance clarity, the key innovations of the above works are summarized in Table 1.

In summary, most research in the construction field has focused on detecting helmet usage, with only a few studies addressing PPE object detection. Currently, no research has been conducted on target recognition of hazards at construction sites. Therefore, this paper proposes a construction site hazard identification system based on YOLOv5n, which includes two detection models: one for unsafe human behavior and one for unsafe conditions of objects. The algorithm enhances YOLOv5n with the ECA [23] attention mechanism, employs SIoU loss [24], EIoU–NMS, and BiFPN [25] to improve detection accuracy, and uses the ghost module [26] to make the model lightweight. Finally, the model is deployed on the Jetson Nano B01 SUB embedded deep learning platform to build the construction site hazard detection system. The contributions of this research can be summarized as follows:

1. Two separate datasets were created: one for unsafe human behavior and one for unsafe conditions of objects. The dataset for unsafe human behavior is the most comprehensive in terms of types of unsafe behavior, including various aspects of helmet usage. Additionally, the first dataset for unsafe conditions of objects on construction sites was established.

2. This study not only applied four attention mechanisms—CA [27], CBAM [28], ECA, and SE [29]—using three different methods but also utilized the ghost module in three different approaches. Finally, three methods that combine attention modules with the ghost model are proposed, laying a foundation for future related research.

3. This study deployed the model on the Jetson Nano B01 SUB embedded deep learning platform to build a construction site hazard detection system. Practical application tests demonstrated that the system operates stably and effectively detects hazards at construction sites.

## Method

### YOLOv5

YOLOv5 is currently one of the most advanced single-stage object detection algorithms. It was released on June 10, 2020, and continues to receive updates. With ongoing iterations, its stability has been widely recognized, especially in construction site recognition. Although newer versions of YOLO (such as YOLOv6–v12) have been released, their improvements have focused mainly on expanding visual tasks—such as segmentation, pose estimation, and tracking—while enhancements in model architecture and object detection accuracy remain limited. Moreover, as functionality becomes richer, overall packaging becomes more complex, increasing the difficulty of migration, deployment, and feature development. Therefore, after comprehensively considering the detection requirements, accuracy and speed, model stability, and scalability, this study selects the YOLOv5 model as the worker recognition model. The network structure of YOLOv5 is shown in Fig 1.

### Attention mechanism

**CA.** Channel attention mechanisms, such as SE attention, have been shown to greatly improve the performance of deep learning models. However, these mechanisms frequently overlook positional information, which is essential for creating spatially selective attention maps. To address this gap, Hou et al. introduced a novel and efficient attention mechanism called CA. This mechanism integrates both horizontal and vertical spatial features into channel attention, enabling the model to capture a wider array of spatial positional information without significantly increasing computational costs. The structure of the CA is illustrated in Fig 2.

**CBAM.** CBAM is a straightforward yet powerful attention module designed for feedforward convolutional neural networks. For a given intermediate feature map, the CBAM module sequentially generates attention maps for both

**Table 1. Summary of different methods from the latest literature.**

| Ref. | Datasets | Method | Originality | Results | Limitations |
|---|---|---|---|---|---|
| [12] | Self-built | YOLOv5s | The ghost module is introduced to reduce model complexity, and the CA and C3CBAM attention modules are added to improve detection accuracy. | Parameters:4.28M GFLOPs:9.2 mAP:93.6% | There are cases of missed measurements and incorrect measurements. |
| [13] | Helmet Detection and SHWD | YOLOv5s | A CA attention module is added to improve detection accuracy, a PSM-EMA optimization strategy is proposed based on positive sample matching and the exponential moving average, and the model's false-positive rate is reduced. | mAP:91.02% | There are cases where small targets at long distances are missed. |
| [14] | Self-built | YOLOv5 | A brand new dataset of heavy equipment at construction sites has been established, which can identify and analyze other standards, including dangerous weather conditions. | mAP:83%(P-PE),93%(Heavy equipment) FPS:141 | The detection capability is poor for small objects. |
| [15] | open-source datasets | YOLOv5s | The performance of the three loss functions, GIoU, DIoU, and CIoU, was verified through experimental comparisons, and the Mosaic-9 data augmentation algorithm was improved. | Precision:93.13% Recall:88.96% | There are cases of missed detections for small objects at long distances. |
| [16] | Self-built | YOLOv5s | A small-object detection layer and MSCA mechanism have been added to improve small-object detection capabilities. The Mish activation function and CIoU loss are used to improve bounding box regression accuracy and accelerate prediction convergence. DIoU–NMS is used to enhance the filtering capability of prediction boxes for occluded small objects. | Recall:94.2% mAP:95.1% | This approach requires more hardware computational resources. |
| [17] | CHV | YOLOv5s | The DilateFormer attention mechanism is used to improve the detection performance of the model; a small target detection layer is added to improve small target detection capabilities. | Parameters:6.09M GFLOPs:15.4 mAP@0.5:93.7% FPS:45 | The model suffers from insufficient generalization capability, while both its computational complexity and inference time have increased. |
| [18] | Self-built | YOLOv5n YOLOv5x | SEIoU loss was proposed to improve the overall performance of the model. Through experimental comparisons, the performance of the four attention mechanisms, ECA, SE, CBAM, and SE, was verified, and CA was ultimately adopted. | Precision:85.5% Recall:80.0% mAP:86.0% | When detecting deformable and easily occluded objects, the model tends to miss detections and produce false-positives. |
| [19] | GDUT-HWD | YOLOv5s | A new hybrid model combining DSConv and SE was proposed to improve feature extraction capabilities; the ghost module was used to lighten the model; and the BAM attention mechanism was combined to reduce the false-negative rate. | Parameters:5.46M GFLOPs:12.6 mAP@0.5:89.1% FPS:72.5 Weight:11.5MB | The detection capability is low for small objects. |
| [20] | SHWD | YOLOv5s | Using ghost convolution reduces the complexity of local models. A new module, CGMS, is proposed to further reduce the computational cost of the model. SE and EIoU loss are adopted to improve the detection performance of the model. | Parameters:2.7M GFLOPs:5.8 Weight:5.8MB Precision:93.4% Recall:86.5% mAP@0.5:92.5% | The model lacks generalization ability. |
| [21] | SHWD | YOLOv5s | The use of the BiFPN improved the overall performance of the model; the introduction of the SE attention mechanism improved detection accuracy; the use of focal EIoU loss improved the convergence of the model; and the use of Soft-NMS improved the accuracy of the model. | Precision:86.5% Recall:77.9% mAP@0.5:82.8% mAP@0.5:0.95:57.7% FPS:83.8 | This approach lacks generalization ability. |
| [22] | augmented SHWD and SBDD | YOLOv5s | SPD-Conv and GAM are integrated to improve the detection accuracy of the model, whereas a decoupled head is used to enhance the model's ability to locate small targets and accelerate its convergence. | F1:93.09% Recall:92.9% mAP@0.5:95.73% mAP@0.5:0.95:61.28% FPS:78.31 Model Size:35.6 | The model has a weak ability to adapt to complex environments. |

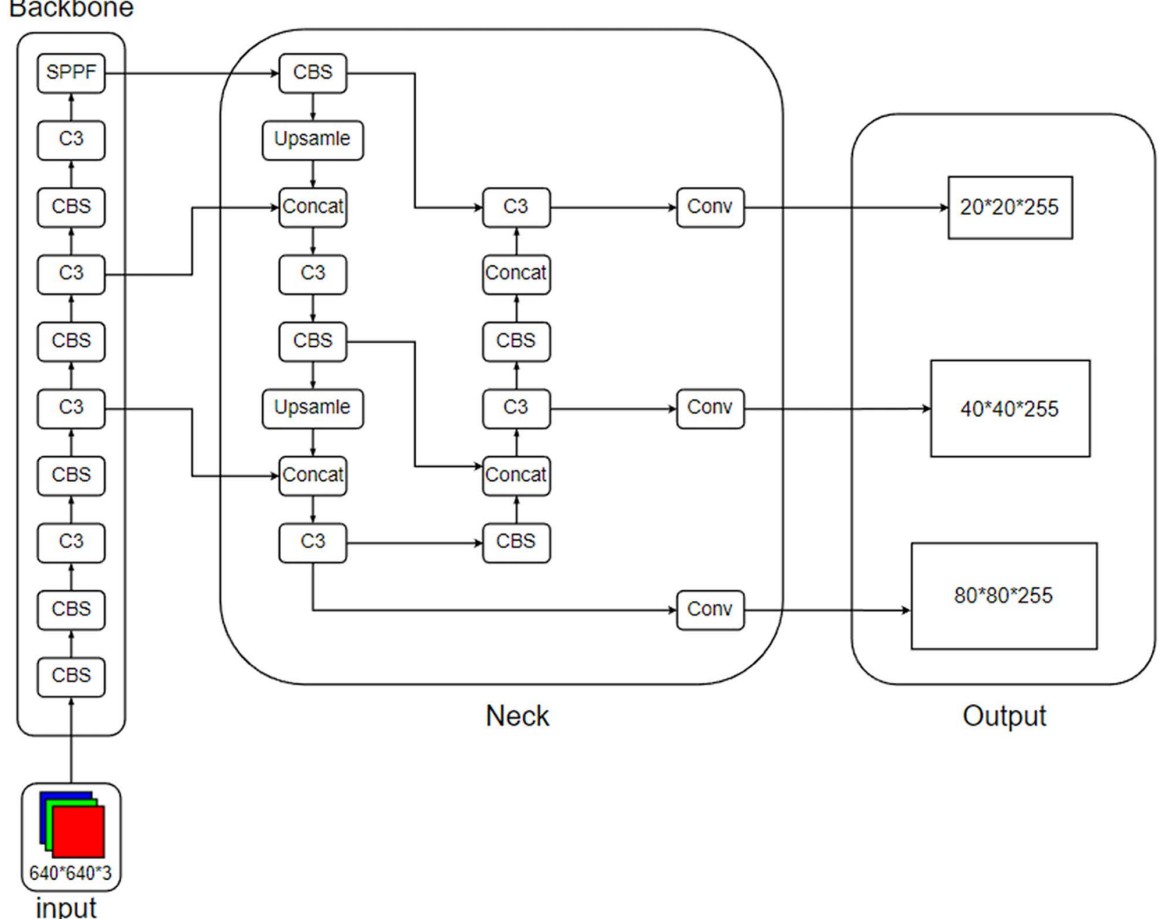

**Fig 1. YOLOv5 network architecture.**

the channel and spatial dimensions. These attention maps are then multiplied with the input feature map to adaptively optimize the features. As a lightweight addition, CBAM seamlessly integrates existing components without requiring extra training, and it has a minimal impact on the detection time. The structure of the CBAM is depicted in Fig 3.

**ECA.** ECA, proposed by Wang et al., is a lightweight module designed to balance performance and model complexity. It achieves significant performance improvement with a minimal increase in parameters. ECA generates channel attention through efficient 1D convolution operations, with the kernel size adaptively determined by a nonlinear mapping of the channel dimension. Although ECA focuses primarily on extracting interchannel information and may not handle spatial information as effectively, it provides an efficient way to enhance feature representation, particularly in scenarios where maintaining computational efficiency while improving model performance is crucial. The ECA structure is shown in Fig 4.

**SE.** SE, introduced by Hu et al., is an innovative architectural unit designed to focus on channel attention. The SE module enhances network expressiveness and performance through its squeeze and excitation operations, which adaptively determine the weights of each channel. This mechanism allows the network to better understand and leverage the relationships between feature channels. By automatically learning the significance and weights of each channel, the SE module enables the network to prioritize crucial feature channels in subsequent layers. Consequently, this improves feature discrimination and overall model performance. The SE structure is illustrated in Fig 5.

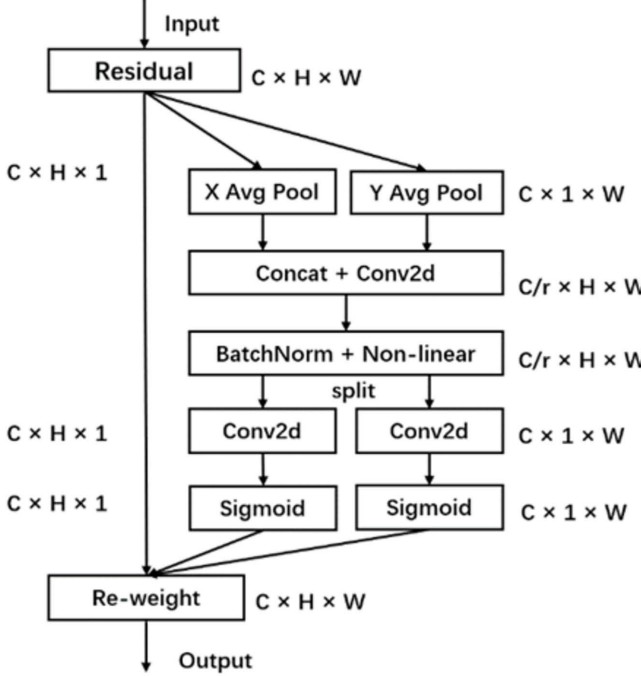

**Fig 2. CA attention mechanism.**

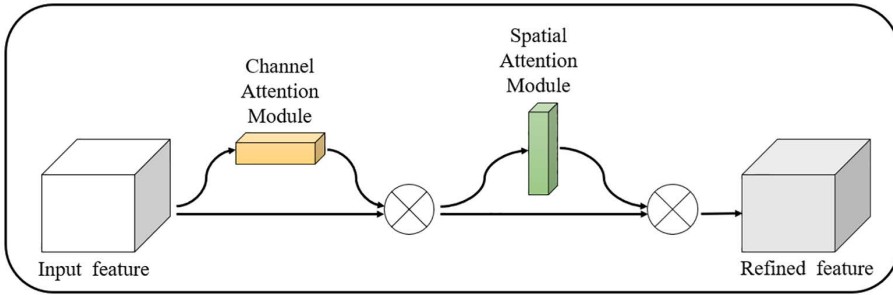

**Fig 3. CBAM attention mechanism.**

### Ghost

The ghost module is a method for creating lightweight neural networks, enabling deep networks to be deployed on computationally weaker mobile devices (such as Jetson Nano) while maintaining performance. It focuses on reducing network model parameters and FLOPs by replacing some convolution operations with inexpensive linear transformations to generate feature maps. The network structure of the ghost module is shown in Fig 6.

### BiFPN

YOLOv5 uses an FPN + PAN structure for feature extraction, as shown in Figs 7(a) and 7(b). However, the PAN does not pay equal attention to features of different scales. Therefore, BiFPN is used to replace PAN and enhance feature extraction capabilities.

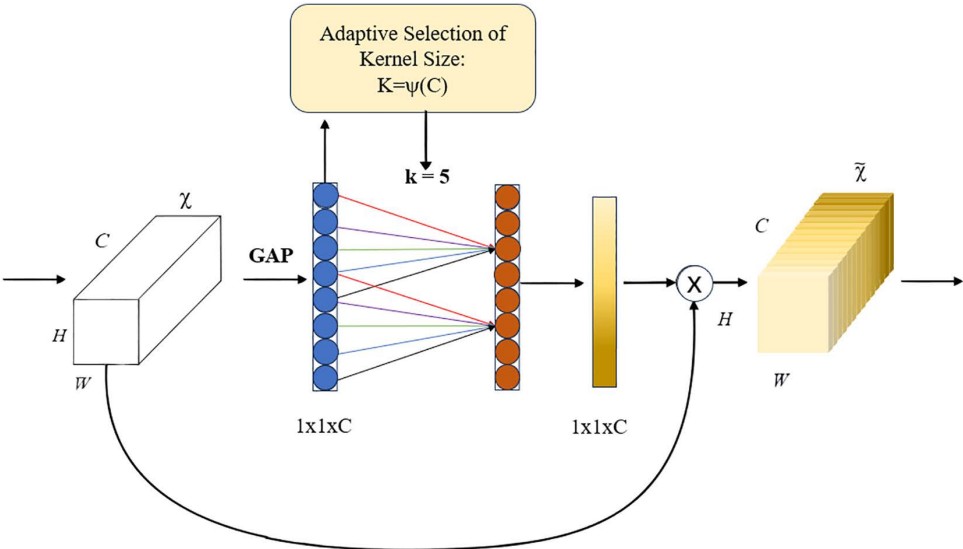

**Fig 4. ECA attention mechanism.**

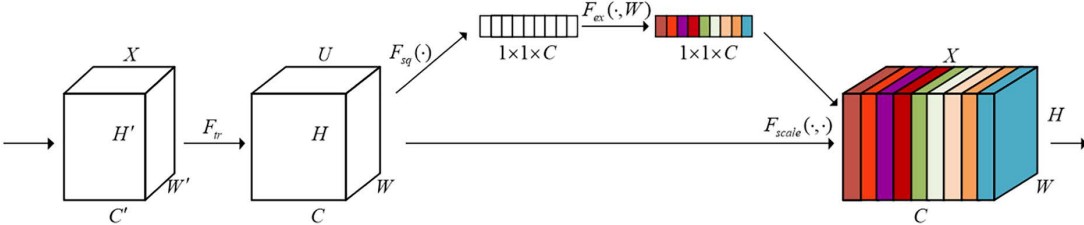

**Fig 5. SE attention mechanism.**

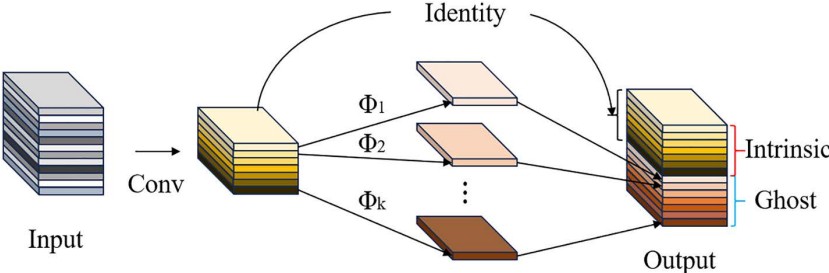

**Fig 6. Ghost module network structure.**

BiFPN, developed by the Google Brain team, is an innovative feature fusion method. As shown in Fig 7(c), BiFPN uses a sophisticated weighted bidirectional feature pyramid structure that optimizes some inefficient nodes present in the PAN, which have only single input and output connections. The improvement in BiFPN is its introduction of cross-layer connections between the input and output nodes at the same scale, enabling richer feature integration without significantly increasing the computational cost. Additionally, BiFPN incorporates a feature fusion module with self-learned weights, featuring bidirectional paths both

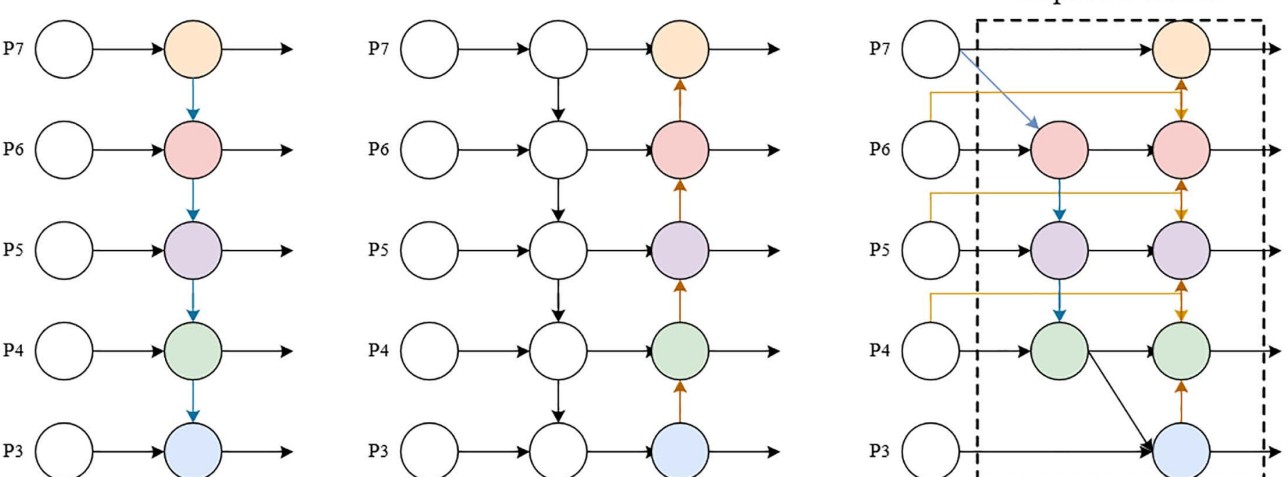

**Fig 7. FPN, PAN, and BiFPN structures.**

top-down and bottom-up. This structure allows BiFPN to achieve higher-level feature aggregation through repeatedly stacked blocks within each module, thereby enhancing feature representation and overall model performance.

## Loss function

With respect to the choice of loss functions, we refer mainly to mainstream loss functions widely used in the literature, such as EIoU loss and SIoU loss. On this basis, we conducted systematic experimental comparisons to select the loss function best suited for this study. Importantly, no customized modifications were made to the loss functions in this research.

**EIoU loss.** The original CIoU loss did not account for the differences in width and height with respect to confidence, which sometimes hindered the model's convergence. To address this issue, Zhang et al. proposed EIoU loss [30]. The specific formula for EIoU loss is as follows:

$$L_{EIOU} = L_{IOU} + L_{dis} + L_{asp} \tag{1}$$

$$L_{IOU} = 1 - IOU + \frac{\rho^2 \left(b, b^{gt}\right)}{(w^c)^2 + (h^c)^2} \tag{2}$$

$$L_{dis} = \frac{\rho^2 \left(w, w^{gt}\right)}{(w^c)^2} \tag{3}$$

$$L_{asp} = \frac{\rho^2 \left(h, h^{gt}\right)}{(h^c)^2} \tag{4}$$

As shown in Fig 8, the yellow box represents the anchor box, the blue box is the target box, and the green box denotes the minimum enclosing box for A and B, with $w^c$ and $h^c$ being the width and height, respectively, of the enclosing box that covers both boxes. The loss function is divided into three parts: the IoU loss $L_{IoU}$, distance loss $L_{dis}$, and aspect ratio loss $L_{asp}$. This approach retains the advantages of CIoU while reducing the discrepancies in width and height between the target and anchor boxes, resulting in faster convergence and better localization results.

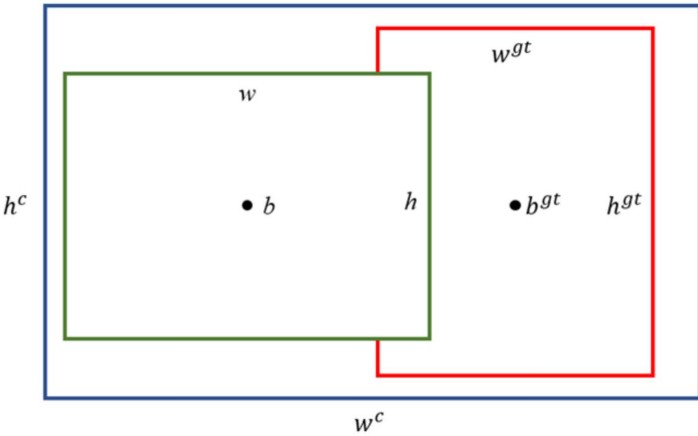

**Fig 8. Schematic diagram of EIOU-loss.**

**SIoU loss.** To address the issue of traditional loss functions not adequately considering the directional differences between the ground truth and predicted bounding boxes, which can slow model convergence, Gevorgyan proposed a new loss function—SIoU loss. SIoU loss extends the original IoU by incorporating the vector angle information between the ground truth and the predicted boxes, providing a more comprehensive measure of their similarity. The SIoU loss design includes four key components that collectively enhance the model's accuracy in directional prediction. Owing to the complexity of the SIoU loss formula, only a simplified version is provided here.

$$L_{SIOU} = 1 - IOU + \frac{\Delta + \Omega}{2} \tag{5}$$

$$\Lambda = 1 - 2 * sin(arcsin(x) - \frac{\pi}{4})) \tag{6}$$

$$\Omega = \sum_{t=w,h} (1 - e^{-\omega_t})^\theta \tag{7}$$

## NMS

**Soft-NMS.** Soft-NMS [31] penalizes and decays the scores with two decay methods. The first penalty function, as shown in Formula (8), is discontinuous, which can lead to abrupt changes in the detection sequence. The continuous penalty function imposes no penalty when there is no overlap and a high penalty when there is significant overlap. To address this, a second Gaussian penalty function is proposed, as shown in Formula (9). This function gradually increases the penalty with lower overlap, thereby avoiding issues related to setting the threshold size in Soft-NMS.

$$s_i = \begin{cases} s_i, & IOU(M, b_i) < N_t \\ 0, & IOU(M, b_i) \geq N_t \end{cases} \tag{8}$$

$$s_i = \begin{cases} s_i, & IOU(M, b_i) < N_t \\ s_i(1 - IOU(M, b_i)), & IOU(M, b_i) \geq N_t \end{cases} \tag{9}$$

$$s_i = s_i e^{-\frac{IOU(M,b_i)^2}{\sigma}}, \forall b_i \notin D \qquad (10)$$

**DIoU–NMS, EIoU–NMS and SIoU–NMS.** In the original NMS, the IoU metric is used to suppress redundant detection boxes, but it often leads to incorrect suppression, as it considers only the overlap area. Therefore, other IoUs, such as EIoU, can be used as NMS criteria. For example, EIoU–NMS, as shown in Formula (15), considers not only the distance between center points but also the differences in width and height along with their confidence. Since each dataset may benefit from different computation methods, this study employs various methods, including EIoU–NMS and SIoU–NMS, which have not been previously explored by other researchers.

$$s_i = \begin{cases} s_i, & EIOU\,(M, b_i) < N_t \\ 0, & EIOU\,(M, b_i) \geq N_t \end{cases} \qquad (11)$$

## Dataset description

The dataset constructed in this study was sourced from publicly available resources, including images from Google, Baidu, and open datasets. After manual annotation, the final dataset was formed. The data collection and analysis methods were in compliance with the terms and conditions of the data sources.

## Experimentation and analysis

### Dataset

This paper divides the targets to be detected into two categories: unsafe human behavior and unsafe object states. However, when relevant images are collected, we find that there are currently no target detection studies on unsafe object states, and even on the internet, only a very small number of images are available. Therefore, we can only divide the dataset into two independent datasets: unsafe human behavior and unsafe object states.

**Dataset on unsafe human behavior.** This study examines whether workers wear safety helmets, use safety ropes, wear reflective vests, and smoke as the subjects of detection. Unlike previous studies, this study categorizes the way safety helmets are worn into two scenarios: proper wearing of safety helmets and improper wearing of safety helmets. Specifically, wearing a helmet with the chin strap fastened is classified as proper wearing, whereas wearing a helmet without fastening the chin strap is classified as improper wearing.

A total of 6,265 images were collected, including images of large targets, small targets, and densely packed targets. The images were then manually labeled using the Labelimg tool, with the annotation format following the YOLO standard, divided into 7 categories: category numbers 0, 1, 2, 3, 4, 5, and 6, corresponding to correct helmet wearing (helmet), no helmet wearing (head), incorrect helmet wearing (incorrect helmet), wearing a seatbelt (belt), smoking (smoking), wearing a safety vest (vest), and human torso (trunk). The labeling process is shown in Fig 9.

YOLOv5 uses mosaic data augmentation during training to increase the training volume, but its effectiveness is limited when the original dataset is small. Therefore, this study expanded the dataset by adding noise, adjusting brightness, rotating, cropping, translating, and mirroring. The process is illustrated in Fig 10. The expanded dataset now contains 12,530 images, with the distribution detailed in Table 2.

## Dataset on unsafe object states

The dataset labels consisted of five categories, numbered 0, 1, 2, 3, and 4, corresponding to scaffold deformation, net breakage, scaffold cracking, missing net, and scaffold punching, respectively. The distribution is shown in Table 3.

Since the dataset for unsafe conditions of objects is small and not representative, this paper conducts improvement experiments on a larger dataset for unsafe behaviors of people. After the optimal improvement scheme is identified, it is directly applied to the dataset for unsafe conditions of objects.

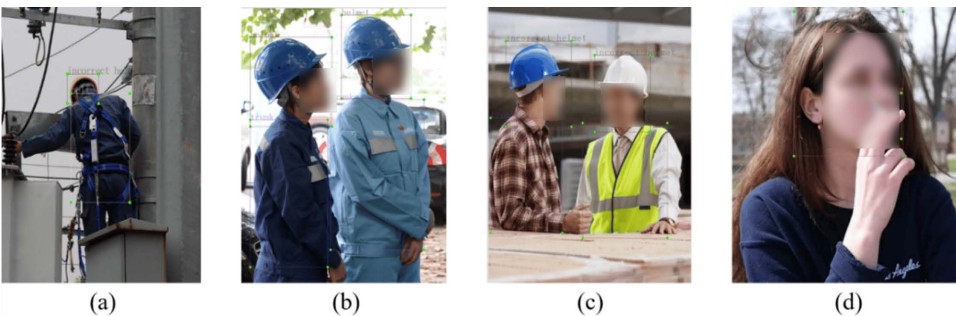

**Fig 9. Image labeling. (a) belt, (b) helmet and trunk, (c) incorrect helmet and vest, and (d) smoking and head.**

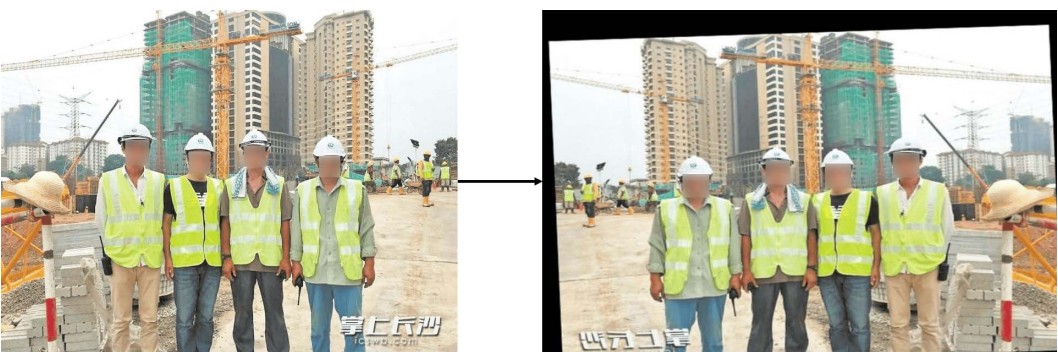

**Fig 10. Data enhancement.**

**Table 2. Human Unsafe Behavior Dataset Distribution.**

| Datasets | Quantity |
|---|---|
| Train | 10024 |
| Val | 2506 |

## Experimental environment

The experimental computers used in this study are equipped with Windows 10, an AMD Ryzen 5 3600 CPU, a GeForce RTX 2070 GPU, 8 GB of video memory, and 32 GB of RAM. All the network models are based on PyTorch 1.10.1 and accelerated using Cuda 11.3 and CuDNN 8.2.1 for GPU acceleration. The image size is 640*640, the number of epochs is 150, the batch size is 24, the warmup is 10, the initial learning rate is 0.01, and the optimization algorithm is SGD.

## Evaluation criteria

The proportion of all detection frames labeled as targets by the model that are actually true targets.

$$Precision = \frac{TP}{TP + FP}$$

(12)

The proportion of all actual targets that are correctly detected by the model.

**Table 3. Insecure state data set distribution of objects.**

| Datasets | Quantity |
|----------|----------|
| Train | 572 |
| Val | 110 |

$$Recall = \frac{TP}{TP + FN} \tag{13}$$

The single-class accuracy (AP) is the average of all possible values of the recall rates. mAP@0.5 is the average accuracy of all classes at an IoU threshold of 0.5, mAP@0.5:0.95 is the average accuracy of all classes at an IoU threshold from 0.5–0.95, with a step size of 0.05, and m is the total number of categories.

$$AP = \int_0^1 P(r)\,dr \tag{14}$$

$$mAP = \frac{1}{m}\sum_{i=1}^{m} AP_i \tag{15}$$

The F1 score (F1) is the harmonic mean of precision and recall, combining the performance of precision and recall.

$$F1 = \frac{2 * Precision * Recall}{Precision + Recall} \tag{16}$$

**Improved YOLOv5**

**Attention mechanism.** There are three ways to improve the attention mechanism: the first is to add the attention module before the SPPF module in the backbone part, the second is to replace the C3 module in the backbone part with the C3 attention module, and the third is to use a mixture of the first two ways of improvement. Taking the CA attention mechanism as an example, the improved structural method is shown in Fig 11; in this part, this study adds the four most commonly used attention mechanisms, CA, CBAM, ECA, and SE, using three methods.

To compare the performance of different attention mechanism modules and ensure the reliability of the comparison, this study placed each attention mechanism module at the same position in the network structure while keeping other parts of the network unchanged. The results for the first improvement method are shown in Table 4. The results for the second improvement method are shown in Table 5. The results for the third improvement method are shown in Table 6.

The results indicate that all the models achieved improvements in detection accuracy after the enhancements. Specifically, CBAM achieves the highest precision and F1 score, which are 0.5% and 0.7% greater than those of YOLOv5n, respectively. SE demonstrated the highest recall and mAP@0.5:0.95, with increases of 1.3% and 0.3% over YOLOv5n, respectively. The ECA recorded the highest mAP@0.5, with a 0.7% improvement over YOLOv5n.

However, all the enhancements increased the model complexity without affecting the number of GFLOPs. Among the attention mechanisms, ECA had the smallest increase in parameters and model size (3 and 1 KB) and only a 4.05% decrease in FPS. Considering these factors, the ECA achieved the best overall performance under the first improvement method.

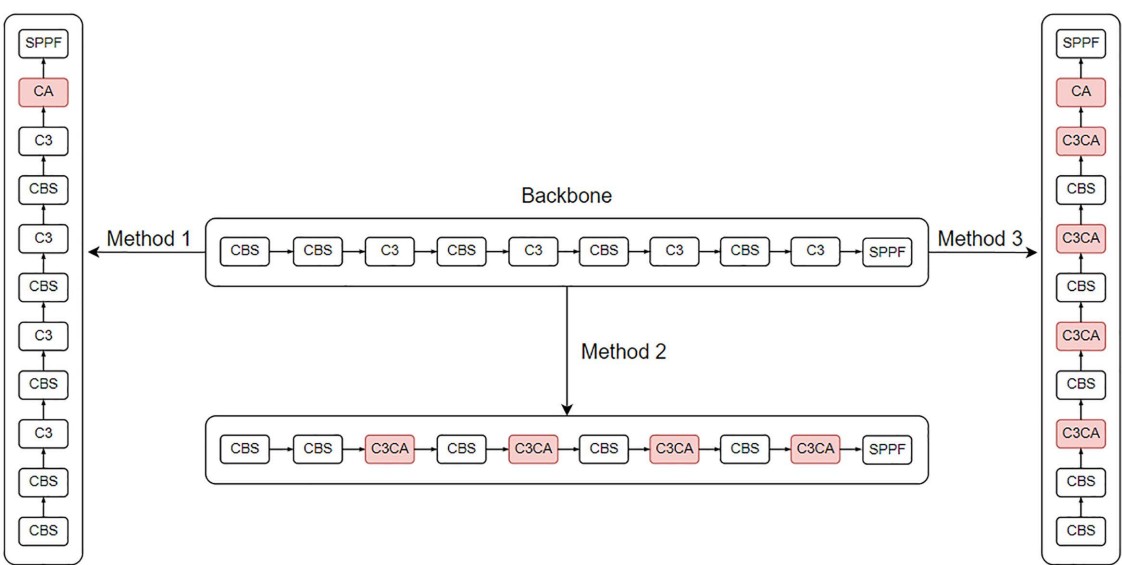

**Fig 11. Improved attention module.**

**Table 4. Experimental results of the first improvement of the attention mechanism.**

| Model | Precision% | Recall% | F1% | mAP@0.5% | mAP@0.5:0.95% | parameters | GFLOPs | FPS | Model size KB |
|---|---|---|---|---|---|---|---|---|---|
| YOLOv5n | 94.4 | 91.8 | 93.1 | 95.7 | 63.1 | 1768636 | 4.2 | 92.995 | 3758 |
| CA | 93.9 | 91.9 | 92.9 | 96.2 | 63.3 | 1775316 | 4.2 | 90.533 | 3776 |
| CBAM | 94.9 | 92.7 | 93.8 | 96.2 | 63.3 | 1776926 | 4.2 | 91.049 | 3778 |
| ECA | 93.7 | 92.9 | 93.3 | 96.4 | 63.2 | 1768639 | 4.2 | 92.618 | 3759 |
| SE | 93.1 | 93.1 | 93.1 | 96.3 | 63.4 | 1776628 | 4.2 | 92.209 | 3776 |

After the second improvement method, adding SE resulted in a decrease in all model performance metrics, with a particularly significant 443% increase in the number of GFLOPs. The other three attention mechanisms improved the model's detection accuracy metrics, with all the indicators exceeding those of the first improvement method.

Among them, CBAM achieved the highest precision, with a 0.1% improvement over YOLOv5n. Compared with YOLOv5n, CA had the highest recall, an improvement of 1.2%. The ECA recorded the highest F1, mAP@0.5 and mAP@0.5:0.95, with increases of 0.4%, 0.9% and 0.7% over YOLOv5n, respectively.

In terms of model complexity, all improvements in the second method reduced model complexity compared with the first method. Specifically, the ECA showed reductions of 8.75% in the parameters, 11.91% in the GFLOPs, and 7.93% in the model size, with only a 4.75% decrease in the FPS. Considering these aspects, the ECA also demonstrated the best overall performance under the second improvement method.

After the third improvement method, all the models exhibited a decrease in detection accuracy metrics. In terms of model complexity, the ECA achieves reductions of 8.77% in parameters, 11.91% in GFLOPs, and 8.44% in model size, with only a 7.06% decrease in FPS. Considering these results, the third improvement method is not suitable for this dataset.

Overall, the second improvement method is more suitable for this dataset. Although CBAM and CA offer higher precision and recall, respectively, ECA provides a higher mAP, and its lower model complexity and higher FPS make it the preferred choice. Therefore, this study opts for the second attention mechanism improvement method.

**Table 5. Experimental results of the second improvement of the attention mechanism.**

| Model | Precision% | Recall% | F1% | mAP@0.5% | mAP@0.5:0.95% | parameters | GFLOPs | FPS | Model size KB |
|---|---|---|---|---|---|---|---|---|---|
| YOLOv5n | 94.4 | 91.8 | 93.1 | 95.7 | 63.1 | 1768636 | 4.2 | 92.995 | 3758 |
| CA | 94 | 93 | 93.5 | 96.4 | 63.7 | 1624124 | 3.7 | 71.306 | 3506 |
| CBAM | 94.5 | 92.2 | 93.3 | 96.5 | 63 | 1618114 | 3.7 | 69.419 | 3482 |
| ECA | 94.2 | 92.8 | 93.5 | 96.6 | 63.8 | 1613577 | 3.7 | 88.582 | 3460 |
| SE | 88.2 | 89.7 | 88.9 | 94.3 | 59.9 | 1622276 | 18.6 | 68.392 | 3760 |

**Table 6. Experimental results of the third improvement of attention mechanisms.**

| Model | Precision% | Recall% | F1% | mAP@0.5% | mAP@0.5:0.95% | parameters | GFLOPs | FPS | Model size KB |
|---|---|---|---|---|---|---|---|---|---|
| YOLOv5n | 94.4 | 91.8 | 93.1 | 95.7 | 63.1 | 1768636 | 4.2 | 92.995 | 3758 |
| CA | 92 | 90 | 91 | 94.7 | 59.8 | 1630804 | 3.7 | 69.036 | 3530 |
| CBAM | 92 | 89.9 | 91 | 94.4 | 59.7 | 1626404 | 3.7 | 68.803 | 3480 |
| ECA | 91.1 | 91.1 | 91.1 | 94.8 | 59.6 | 1613580 | 3.7 | 86.425 | 3441 |
| SE | 88.5 | 87 | 87.8 | 92.2 | 56.4 | 1630468 | 18.6 | 68.421 | 3777 |

**Ghost lightweight improvements.** There are three types of ghost lightweight improvements: the first replaces the Conv module (referred to as the Conv module in this section) in the CBS with a GhostConv module, and the second replaces the C3 module with a C3Ghost module, which is different from the attention mechanism in that the ghost module can generally replace both the Conv module and the C3 module in the neck section. The third is a mixture of the first two improvements. The improvement of the first method is shown in Fig 12, the second method is shown in Fig 13, and the third method is shown in Fig 14. In this paper, the CBS structure in the figure is replaced by the Conv module for easy observation.

To determine the best way to improve the force, a total of three different sets of improvement experiments are performed in this paper. The results of the experiments are shown in Table 7, where GhostConv is the first improvement method, C3Ghost is the second improvement method, ghost is the third improvement method, and FPS-j is the FPS data of the model on Jetson Nano B01.

From the table, it is evident that as more ghost modules are added to the network structure, both the detection accuracy metrics and model complexity decrease, while FPS decreases unusually, but FPS-j increases. This is because ghost modules are optimized for devices with lower computational capabilities, and these optimizations can hinder performance on high-end GPUs. Although the precision, recall, F1, mAP@0.5, and mAP@0.5:0.95 metrics of the ghost model are the lowest, it achieves the lowest model complexity, with parameters, GFLOPs, and model size reduced by 46.51%, 45.24%, and 41.38%, respectively, and FPS-j improved by 23.41%. Considering that the model will be deployed on more accessible mobile devices such as Jetson Nano, this study adopts the third improvement method.

**Mixed use of ghost and attention modules.** A review of the above experiments reveals that the improvement of the ghost module and the improvement of the attention mechanism conflict with each other, i.e., the C3 module can be replaced only by the C3ECA module or the C3Ghost module. Therefore, in this work, to find the best fusion improvement method, three groups of experiments were designed, each containing three subexperiments. For the reliability of the experiments, CA and CBAM, which are close to the performance of the ECA, are also added to the experiments in this paper. The first set of experiments adds the attention mechanism to the backbone part before the SPPF module, which is based on the improvement of using the ghost module in the third improvement method, as shown in Fig 15 (taking CA as an example). The second set of experiments consisted of replacing the C3 module in the backbone section with the C3 attention module and then replacing the remaining C3 module with the ghost module, as shown in Fig 16 (using CA as an

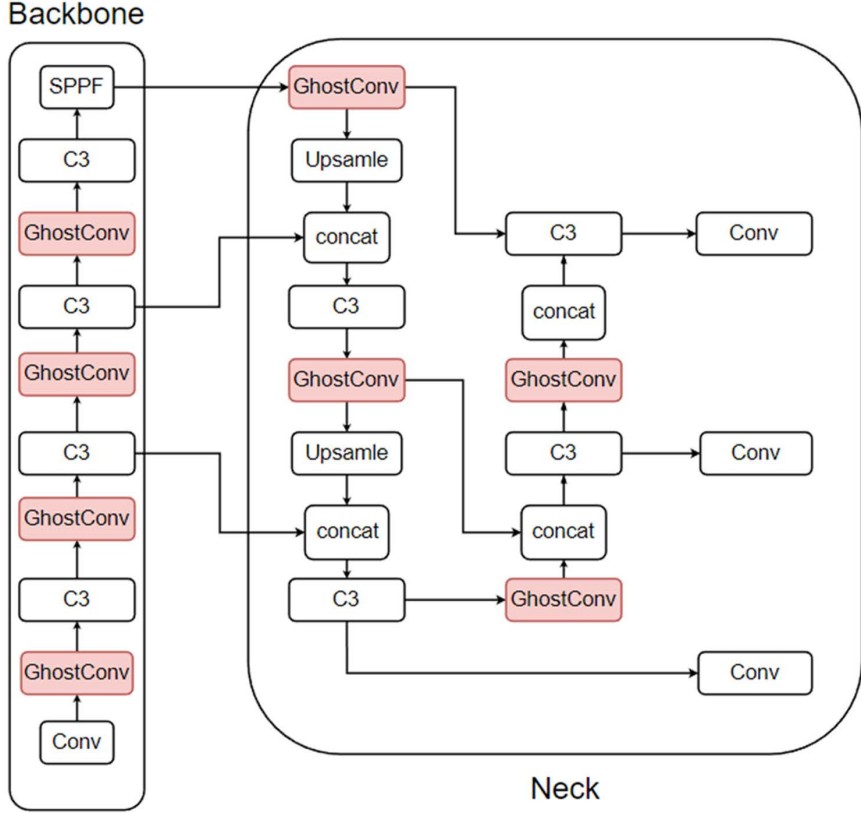

**Fig 12. Conv module replaced with GhostConv module.**

example). The third set of experiments adds the attention mechanism to the backbone part before the SPPF module and replaces the C3 module in the backbone part with the C3 attention module and then replaces the remaining C3 module with the ghost module, as shown in Fig 17 (as an example of CA). The results of the first set of experiments are shown in Table 8, the results of the second set of experiments are shown in Table 9, and the results of the third set of experiments are shown in Table 10.

Adding attention modules to the backbone section resulted in a model performance that was even lower than that of the ghost model with no attention mechanism. Therefore, applying the attention mechanism to the SPPF module of the backbone before the third ghost improvement method is used does not suit this dataset. Unlike previous experimental results, the ECA performs better than the CA and CBAM methods do in this context.

Table 8 shows that the experimental results obtained by replacing some C3Ghost modules with C3 attention modules are similar to those of the second improvement method for attention mechanisms. Among the three attention mechanisms, CA shows the most significant improvement in precision, with a 1.4% increase compared with the ghost model. Compared with the ghost model, the CBAM shows the most significant improvement in recall, with a 1.8% increase. Compared with the ghost model, the ECA yields the most significant improvements in F1, mAP@0.5 and mAP@0.5:0.95, with increases of 0.9%, 0.7% and 1.7%, respectively. ECA also has the lowest complexity among the three attention mechanisms, with parameters, GFLOPs, and model size improving by 13.94%, 13.04%, and 10.3%, respectively, compared with the ghost model, whereas FPS-j decreases by 22.78%. Importantly, although this method slightly increases the complexity of the model, it effectively improves the model's detection performance.

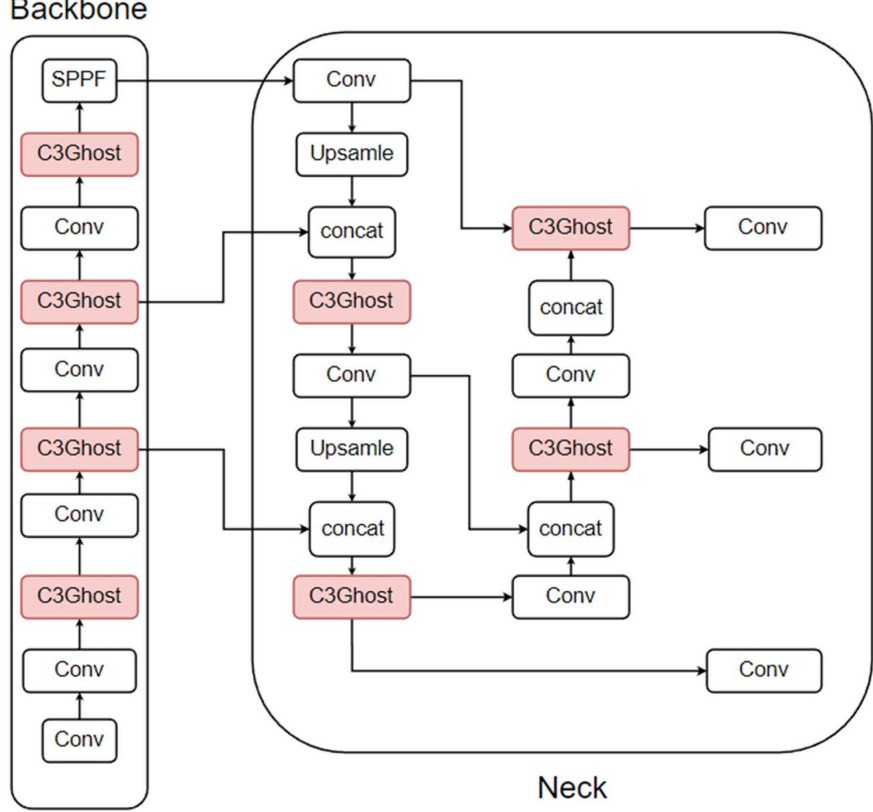

**Fig 13. C3 module replaced with C3Ghost module.**

The data from the second and third groups clearly show that the experimental results of the third group are similar to those of the second group, but all the models in the third group perform worse than those in the second group. This finding indicates that the model's detection accuracy does not necessarily improve with increasing number of attention modules; in fact, too many attention modules may even degrade the model's overall performance.

In summary, the results from the three experiments show that the second group's method, which involves first replacing the C3 modules in the backbone with C3 attention modules and then using ghost modules to replace the remaining C3 modules, performs best on this dataset. Among the attention mechanisms, the ECA remains the most effective, so this method is used in this study to combine ghost modules with ECA attention modules, referred to as C3ECAGhost in the subsequent sections.

**BiFPN.** There are generally two methods for merging BiFPNs: add and concat. The improved network model is shown in Fig 18 (using concat as an example).

Add: The values of two feature maps are added together while keeping the number of image channels unchanged. The advantage of this method is that it reduces the computational load.

Concat: This method involves stacking the channels directly without changing the values within each feature map. Essentially, it combines the two feature maps into one, but the feature maps remain independent of each other.

For the reasonableness and reliability of the experiment, 2 sets of experiments are performed in this paper according to these two improvements, and the experimental results are shown in Table 11.

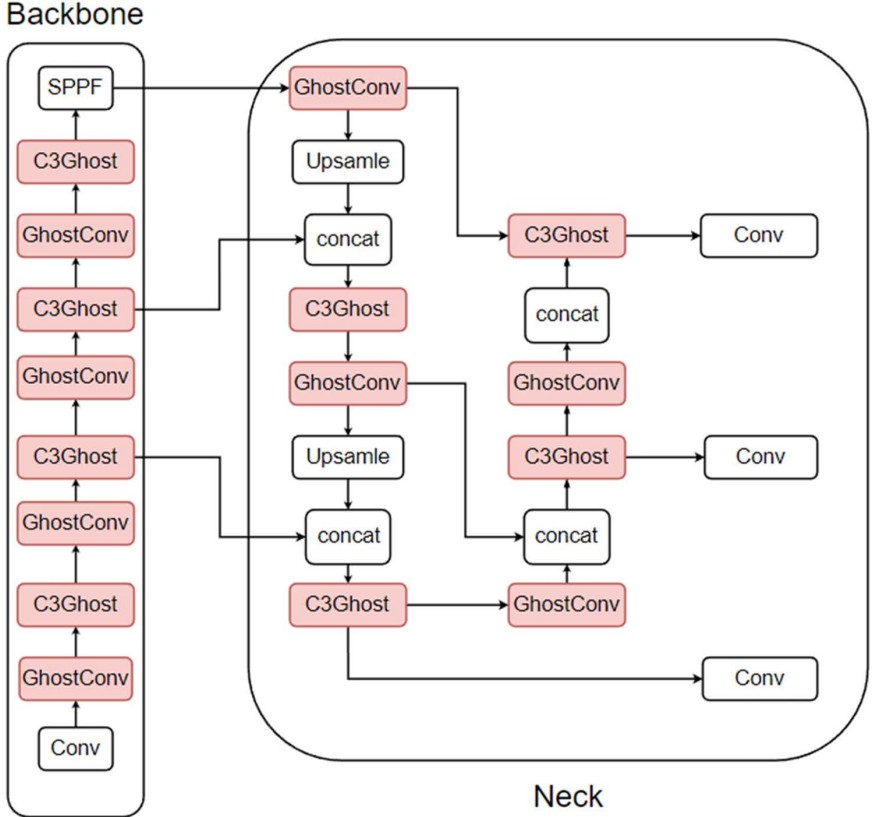

**Fig 14. Mixing the first two improvements.**

**Table 7. Ghost improvement experiment results.**

| Model | Precision% | Recall% | F1% | mAP@0.5% | mAP@0.5:0.95% | parameters | GFLOPs | FPS | FPS-j | Model size KB |
|---|---|---|---|---|---|---|---|---|---|---|
| YOLOv5n | 94.4 | 91.8 | 93.1 | 95.7 | 63.1 | 1768636 | 4.2 | 92.995 | 11.243 | 3758 |
| GhostConv | 93.3 | 90 | 91.6 | 94.9 | 61.1 | 1470956 | 3.6 | 91.943 | 12.217 | 3174 |
| C3Ghost | 91.5 | 90.2 | 90.8 | 94.3 | 59.2 | 1243720 | 2.8 | 87.752 | 13.124 | 2767 |
| Ghost | 91 | 87.3 | 89.1 | 92.6 | 56.4 | 946040 | 2.3 | 83.661 | 13.875 | 2203 |

The table shows that both methods can improve the model's detection performance. Compared with the add operation, the concat operation yields higher precision, recall, F1, mAP@0.5, and mAP@0.5:0.95, with increases of 0.7%, 0.8%, 0.1%, 0.6%, and 2%, respectively. Moreover, the parameters, GFLOPs, and model size increase by 1.34%, 2.33%, and 2%, respectively. Importantly, FPS improves by 6.93%, whereas the addition operation can even decrease the model's FPS. Therefore, this paper uses the concat method for adding BiFPN, referred to as BiFPN, in the following sections.

**Loss function.** To determine the best way to improve the results, a total of 2 sets of different improvement experiments are performed in this paper. The results of the experiments are shown in Table 12.

The study revealed that changing the loss function does not alter the model's parameters or GFLOPs and even results in a 6% reduction in model size. Both loss functions improve the model's detection performance; however, while the precision remains the same for both, the SIoU loss has a 0.4% lower recall and 0.2% lower F1 than the EIoU loss but achieves

Backbone

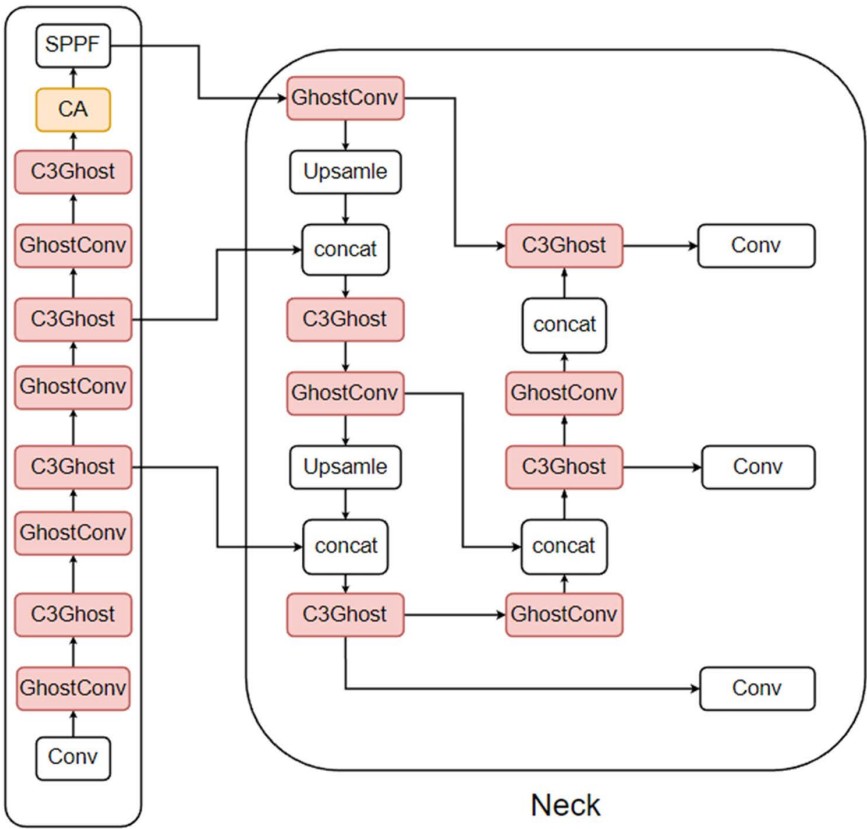

**Fig 15. The way the first group of experiments was improved.**

higher mAP@0.5 and mAP@0.5:0.95, with increases of 0.3% and 1.8%, respectively. Therefore, this paper uses the SIoU loss to increase the model's performance.

**NMS.** To determine the best way to improve the results, a total of 4 sets of different improvement experiments are performed in this paper. The results of the experiments are shown in Table 13.

The table shows that after improving NMS, the model's parameters, GFLOPs, and model size remain unchanged, but FPS slightly increases, demonstrating that NMS improvements can indeed enhance detection efficiency. In terms of detection performance, Soft-NMS shows the most significant increase in mAP@0.5:0.95%, an improvement of 3.2% compared with YOLOv5n. EIoU–NMS shows the most notable increases in precision, recall, F1 and mAP@0.5, with improvements of 0.5%, 2.3%, 1.4% and 1.5%, respectively. Given that mAP@0.5% has greater general applicability, this study adopts EIoU–NMS.

**Ablation experiment.** On the basis of the experiments above, a detailed improvement involving the use of the ECA attention mechanism, ghost module, BiFPN, SIoU loss, and EIoU–NMS to enhance YOLOv5 is presented. This study employs ablation experiments to verify the impact of each combined improvement on model performance. The experimental results are shown in Table 13. For convenience, the improved models are named using the initials of the modifications, such as YOLO-CGBSE, which represents the integration of C3ECAGhost, BiFPN, SIoU loss, and EIoU–NMS (Table 14).

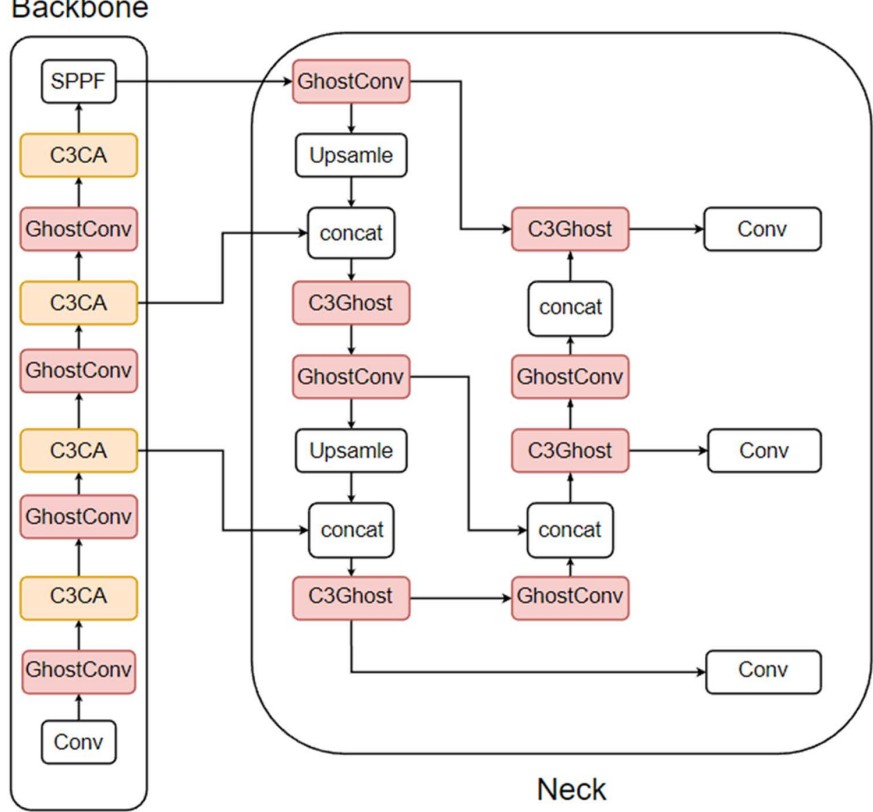

**Fig 16. The way the second group of experiments was improved.**

The impact of each improvement is consistent with the results from the previous experiments. Although the YOLO-CGBSE model has higher parameters, GFLOPs, FPS-j, and model size than the ghost model does, it achieves the highest precision, recall, F1, mAP@0.5%, and mAP@0.5:0.95%, with improvements of 0.5%, 1.9%, 1.2%, 1%, and 2.1% over the ghost model, respectively. To balance both detection accuracy and speed, this paper selects the YOLO-CGBSE model as the model for detecting unsafe behaviors in humans. Overall, the ECA, as a lightweight channel attention mechanism, enhances feature representation, particularly for small object detection, while incurring minimal computational and parameter overhead. Ghost retains key information while significantly reducing the number of parameters and providing a plug-and-play solution with strong versatility. SIoU loss not only optimizes regression loss and accelerates convergence but also significantly improves detection performance in dense scenes and for small objects. EIoU–NMS can more effectively remove overlapping targets, avoid false deletions, and retain more precisely located boxes, thereby improving detection accuracy.

The YOLO-CGBSE network structure is shown in Fig 19, and the P curve, R curve, PR curve, and F1 curve are shown in Fig 20. The pseudocode is shown in Algorithm 1.

---

Algorithm 1 Pseudocode for the YOLO-CGBSE algorithm

---

**Input:** images Img, Bounding box coordinates $B_x$, $B_y$, width $B_w$, height $B_h$
**Output:** Class probabilities $P_c$ and Predicted Bounding box coordinates
1:**Initialize:** Img = Img_train(80%) + Img_val(20%);
2:batch_size = 24;

---

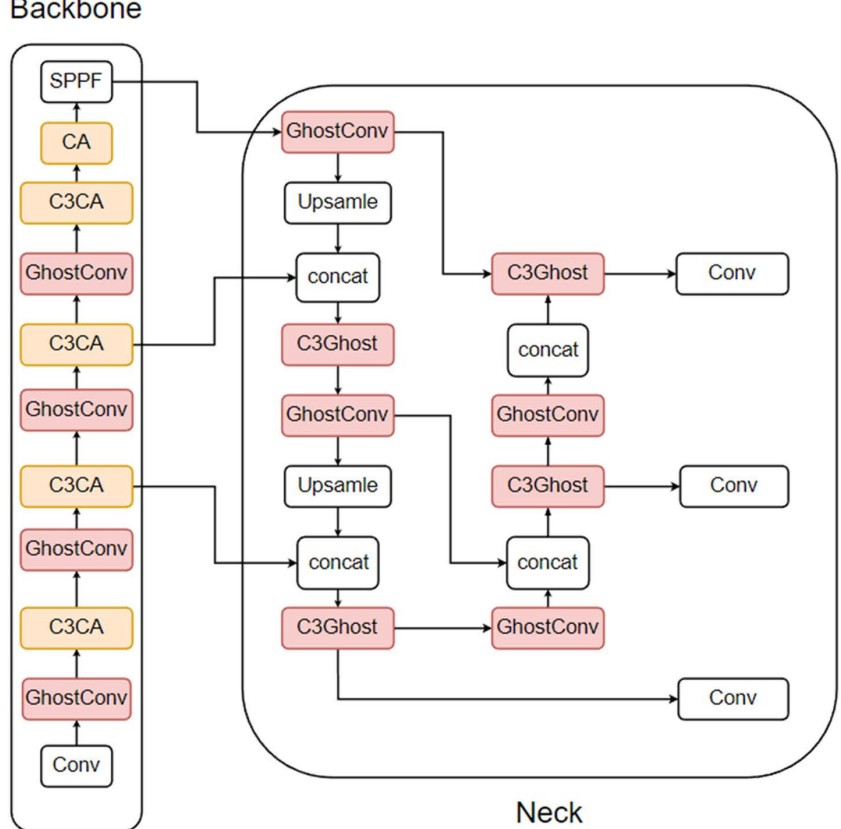

**Fig 17. The way the third group of experiments was improved.**

**Table 8. Results of the first group of experiments.**

| Model | Precision% | Recall% | F1% | mAP@0.5% | mAP@0.5:0.95% | parameters | GFLOPs | FPS | FPS-j | Model size KB |
|-------|-----------|---------|-----|----------|----------------|-----------|--------|-----|-------|---------------|
| YOLOv5n | 94.4 | 91.8 | 93.1 | 95.7 | 63.1 | 1768636 | 4.2 | 92.995 | 11.243 | 3758 |
| Ghost | 91 | 87.3 | 89.1 | 92.6 | 56.4 | 946040 | 2.3 | 83.661 | 13.875 | 2203 |
| CA | 90.1 | 86.6 | 88.3 | 91.7 | 55.4 | 993680 | 2.3 | 80.245 | 10.327 | 2301 |
| CBAM | 89.3 | 85.7 | 87.5 | 90.9 | 54.8 | 995290 | 2.3 | 79.939 | 10.214 | 2302 |
| ECA | 90.4 | 86.9 | 88.6 | 92.1 | 56 | 987003 | 2.3 | 83.148 | 10.755 | 2284 |

**Table 9. Experimental results of the second group.**

| Model | Precision% | Recall% | F1% | mAP@0.5% | mAP@0.5:0.95% | parameters | GFLOPs | FPS | FPS-j | Model size KB |
|-------|-----------|---------|-----|----------|----------------|-----------|--------|-----|-------|---------------|
| YOLOv5n | 94.4 | 91.8 | 93.1 | 95.7 | 63.1 | 1768636 | 4.2 | 92.995 | 11.243 | 3758 |
| Ghost | 91 | 87.3 | 89.1 | 92.6 | 56.4 | 946040 | 2.3 | 83.661 | 13.875 | 2203 |
| CA | 92.4 | 87 | 89.6 | 93.2 | 57.9 | 1088500 | 2.7 | 67.819 | 9.857 | 2746 |
| CBAM | 90.6 | 89.1 | 89.8 | 93.3 | 58 | 1082490 | 2.7 | 68.219 | 9.924 | 2452 |
| ECA | 91.2 | 88.9 | 90 | 93.3 | 58.1 | 1077953 | 2.6 | 82.519 | 10.714 | 2430 |

**Table 10. Results of the third group of experiments.**

| Model | Precision% | Recall% | F1% | mAP@0.5% | mAP@0.5:0.95% | parameters | GFLOPs | FPS | FPS-j | Model size KB |
|---|---|---|---|---|---|---|---|---|---|---|
| YOLOv5n | 94.4 | 91.8 | 93.1 | 95.7 | 63.1 | 1768636 | 4.2 | 92.995 | 11.243 | 3758 |
| Ghost | 91 | 87.3 | 89.1 | 92.6 | 56.4 | 946040 | 2.3 | 83.661 | 13.875 | 2203 |
| CA | 90.7 | 88.5 | 89.6 | 92.9 | 57.5 | 1136140 | 2.7 | 67.89 | 9.616 | 2574 |
| CBAM | 90.6 | 88.9 | 89.7 | 93 | 57.6 | 1131740 | 2.7 | 65.26 | 9.734 | 2551 |
| ECA | 90.7 | 88.7 | 89.7 | 93.3 | 57.3 | 1118916 | 2.7 | 79.862 | 10.415 | 2512 |

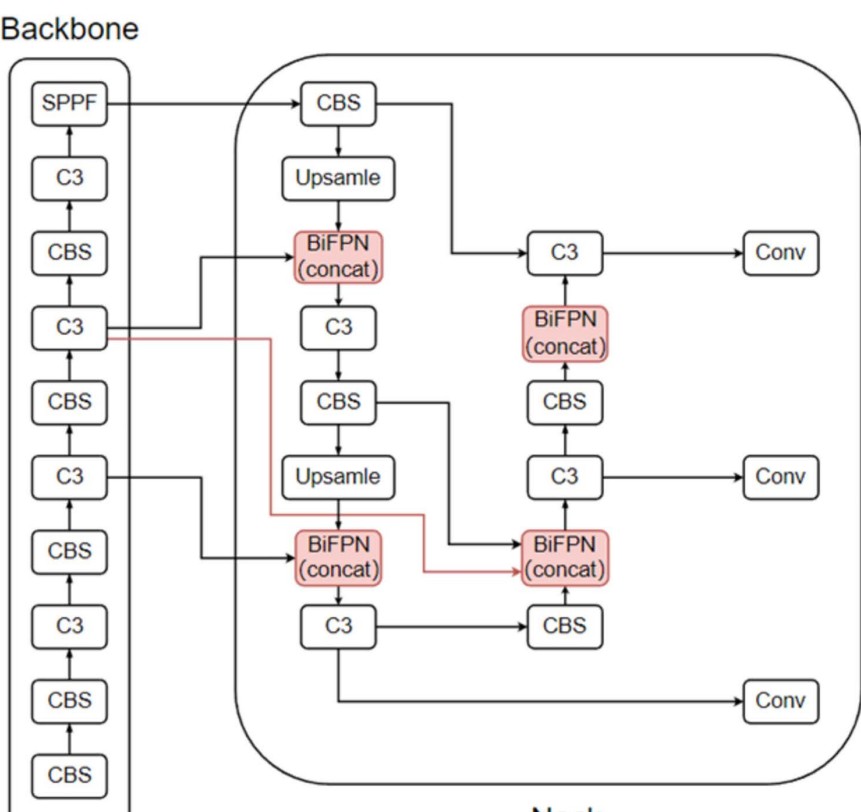

**Fig 18. BiFPN (concat) improvement approach.**

**Table 11. BiFPN improvement experimental results.**

| Model | Precision% | Recall% | F1% | mAP@0.5% | mAP@0.5:0.95% | parameters | GFLOPs | FPS | Model size KB |
|---|---|---|---|---|---|---|---|---|---|
| YOLOv5n | 94.4 | 91.8 | 93.1 | 95.7 | 63.1 | 1768636 | 4.2 | 92.995 | 3758 |
| add | 92.9 | 92.6 | 92.7 | 95.7 | 61.2 | 1806021 | 4.3 | 87.082 | 3851 |
| concat | 93.6 | 92.8 | 93.2 | 96.3 | 63.2 | 1830149 | 4.4 | 93.114 | 3859 |

**Table 12. Experimental results of loss function improvement.**

| Model | Precision% | Recall% | F1% | mAP@0.5% | mAP@0.5:0.95% | parameters | GFLOPs | FPS | Model size KB |
|---|---|---|---|---|---|---|---|---|---|
| YOLOv5n | 94.4 | 91.8 | 93.1 | 95.7 | 63.1 | 1768636 | 4.2 | 92.995 | 3758 |
| EIOU-loss | 94.5 | 92.8 | 93.6 | 96.2 | 63 | 1768636 | 4.2 | 92.861 | 3737 |
| SIOU-loss | 94.5 | 92.4 | 93.4 | 96.5 | 64.8 | 1768636 | 4.2 | 94.79 | 3737 |

**Table 13. Results of NMS improvement experiments.**

| Model | Precision% | Recall% | F1% | mAP@0.5% | mAP@0.5:0.95% | parameters | GFLOPs | FPS | Model size KB |
|---|---|---|---|---|---|---|---|---|---|
| YOLOv5n | 94.4 | 91.8 | 93.1 | 95.7 | 63.1 | 1768636 | 4.2 | 92.995 | 3758 |
| Soft-NMS | 94.8 | 92.9 | 93.8 | 96.7 | 66.3 | 1768636 | 4.2 | 95.906 | 3758 |
| DIOU-NMS | 94.5 | 93.7 | 94.1 | 97 | 65.2 | 1768636 | 4.2 | 96.395 | 3758 |
| EIOU-NMS | 94.9 | 94.1 | 94.5 | 97.2 | 65.4 | 1768636 | 4.2 | 96.532 | 3758 |
| SIOU-NMS | 94.6 | 88.1 | 91.2 | 92.1 | 62.1 | 1768636 | 4.2 | 96.022 | 3758 |

**Table 14. Results of ablation experiments.**

| Model | Precision% | Recall% | F1% | mAP@0.5% | mAP@0.5:0.95% | parameters | GFLOPs | FPS | FPS-j | Model size KB |
|---|---|---|---|---|---|---|---|---|---|---|
| Ghost | 91 | 87.3 | 89.1 | 92.6 | 56.4 | 946040 | 2.3 | 83.661 | 13.875 | 2203 |
| YOLO-CG | 91.2 | 88.9 | 90. | 93.3 | 58.1 | 1077953 | 2.6 | 82.519 | 10.714 | 2430 |
| YOLO-CGB | 91.2 | 89.1 | 90.1 | 93.4 | 58.5 | 1121834 | 2.8 | 78.712 | 10.125 | 2518 |
| YOLO-CGBS | 90.9 | 89.1 | 90 | 93.3 | 58.1 | 1121834 | 2.8 | 79.873 | 10.267 | 2518 |
| YOLO-CGBSE | 91.5 | 89.2 | 90.3 | 93.6 | 58.5 | 1121832 | 2.8 | 78.8 | 10.114 | 2497 |

```
3:epochs E = 200;
4:Generation G = 200;
5:for i = 1:G do
6:   for j = 1 : batch_size do
7:     Load yolo hyperparameters configuration file;
8:     Run Genetic Algorithm (GA) to obtain best hyperparameter values;
9: end for
10:end for
11:Training Stage:
12:for i = 1:E do
13:   Load optimized yolo training parameters from GA;
14:   Training on the Img_train with the YOLO-CGBSE;
15:   Evaluating algorithm using Img_val;
16: end for
17: save optimal checkpoint bestweight.pt
```

## Model comparison

To further validate the performance of YOLO-CGBSE, it is compared with existing mainstream target detection algorithms, and the results are shown in Table 15.

The table shows that although the YOLOv3, YOLOv5s, YOLOv5m, YOLOv5l, and YOLOv5x models perform better in terms of precision, recall, F1, mAP@0.5, and mAP@0.5:0.95 than the YOLO-CGBSE model does, their parameters, GFLOPs, FPS, and model sizes are less acceptable, making these models impractical for deployment. YOLOv7-tiny and YOLOv8n, although slightly superior to YOLO-CGBSE in terms of detection accuracy and FPS, perform poorly on mobile devices and struggle to run smoothly. YOLOv9 falls short of YOLO-CGBSE in all the metrics, with the most notable

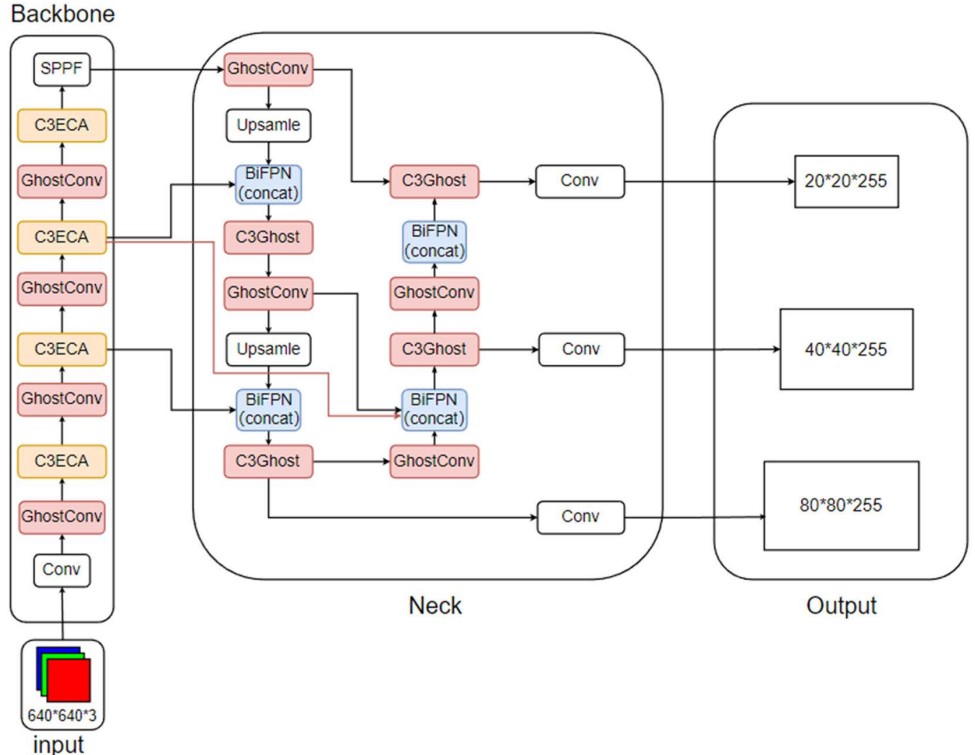

**Fig 19. YOLO-CGBSE network architecture.**

difference being in FPS. As the version number increases, the detection performance of YOLOv10n–YOLOv12n continues to improve, but the model complexity also increases accordingly. More importantly, owing to compatibility issues, YOLOv10n–YOLOv12n cannot run normally on the Jetson Nano B01 device.

YOLO-CGBSE has higher FPS, a smaller model size, and extremely low computing resources, which are clearly more conducive to deployment. To more intuitively visualize the differences between models, this paper provides some characteristic images as detection targets, and the detection results are compared, as shown in Fig 21.

As shown in Fig 21, both the original YOLOv5n and YOLO-CGBSE suffer from misdetection and leakage problems in the UAV view, but the leakage rate of YOLO-CGBSE is lower. When detecting high-resolution images, YOLOv5n suffers from the problem of multiple prediction frames for the same target, which is considered in this paper to be a flaw in the original NMS computation method that leads to this problem; when detecting indoor work images, YOLOv5n incorrectly labels the shaded arm as a helmet that is not correctly worn; when detecting aerial work images, YOLOv5n incorrectly labels the uppermost seatbelt as a torso; when detecting shaded aerial work pictures, YOLOv5n not only detects the construction worker on the left side but also incorrectly labels a torso label as a seatbelt.

## Modeling of unsafe object state detection

After a series of experiments above, the final improvement program is determined. In this section, the detection model trained on the unsafe object state dataset is proposed, and the model is named YOLO-M. The performance of the model is shown in Table 16.

Compared with that of the detection model for unsafe human behavior, the detection performance index of the detection model for unsafe object states is abnormally high, and there is not much difference in the detection performance

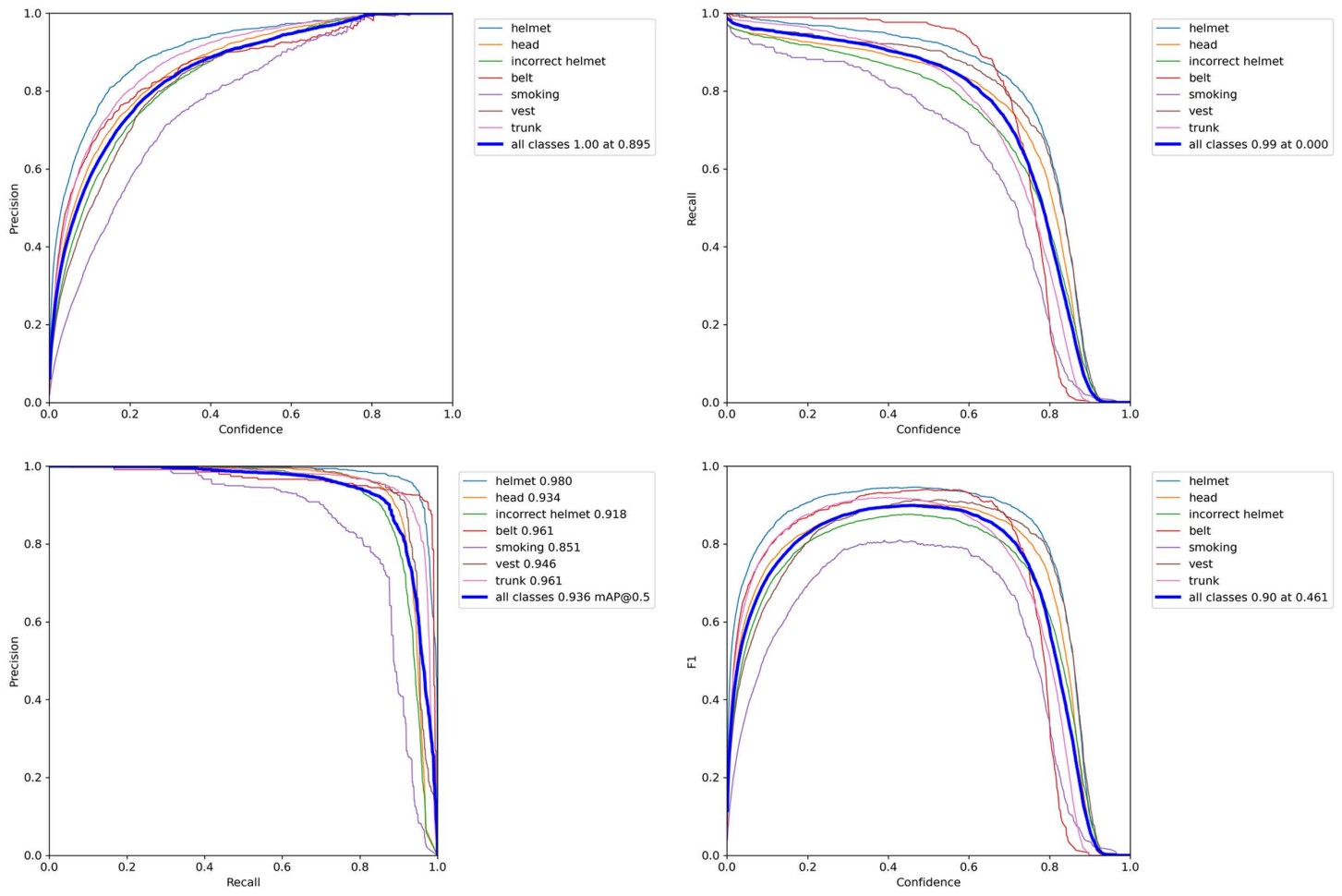

**Fig 20. P curve, R curve, PR curve and F1 curve.**

**Table 15. Model comparison results.**

| Model | Precision% | Recall% | F1% | mAP@0.5% | mAP@0.5:0.95% | parameters | GFLOPs | FPS | FPS-j | Model size KB |
|---|---|---|---|---|---|---|---|---|---|---|
| YOLOv5s | 98 | 96.5 | 97.2 | 98.9 | 74.9 | 7029004 | 15.8 | 78.324 | 5.17 | 14018 |
| YOLOv5m | 99 | 98.1 | 98.5 | 99.3 | 84.6 | 20877180 | 47.9 | 60.79 | 2.79 | 41164 |
| YOLOv5l | 99.4 | 98.5 | 98.9 | 99.5 | 89.5 | 46140588 | 107.7 | 35.723 | 1.68 | 90621 |
| YOLOv5x | 99 | 99.1 | 99 | 99.5 | 92 | 86213788 | 203.9 | 20.66 | 0.98 | 169021 |
| YOLOv3 | 98.8 | 99.3 | 99. | 99.5 | 90.4 | 91529740 | 156.6 | 26.831 | 1.32 | 120595 |
| YOLOv7-tiny | 91.8 | 91.6 | 91.7 | 95.1 | 61.8 | 6031224 | 13.2 | 80.645 | 5.423 | 11996 |
| YOLOv8n | 93.3 | 90.8 | 92. | 95.9 | 70.2 | 3007013 | 8.1 | 79.365 | 7.241 | 6074 |
| YOLOv9t | 89.3 | 84.5 | 86.8 | 90.7 | 59.3 | 2619290 | 10.7 | 31.348 | 2.96 | 5935 |
| YOLOv10n | 91.7 | 89.3 | 90.5 | 94.4 | 67 | 2762608 | 8.6 | 73.529 | × | 5624 |
| YOLOv11n | 91.8 | 89.6 | 90.7 | 94 | 66.3 | 2583517 | 6.3 | 72.463 | × | 5337 |
| YOLOv12n | 93 | 89.4 | 91.1 | 94.1 | 67.3 | 2509709 | 5.8 | 69.93 | × | 5302 |
| YOLO-CGBSE | 91.5 | 89.2 | 90.3 | 93.6 | 58.5 | 1121832 | 2.8 | 78.8 | **10.114** | 2497 |

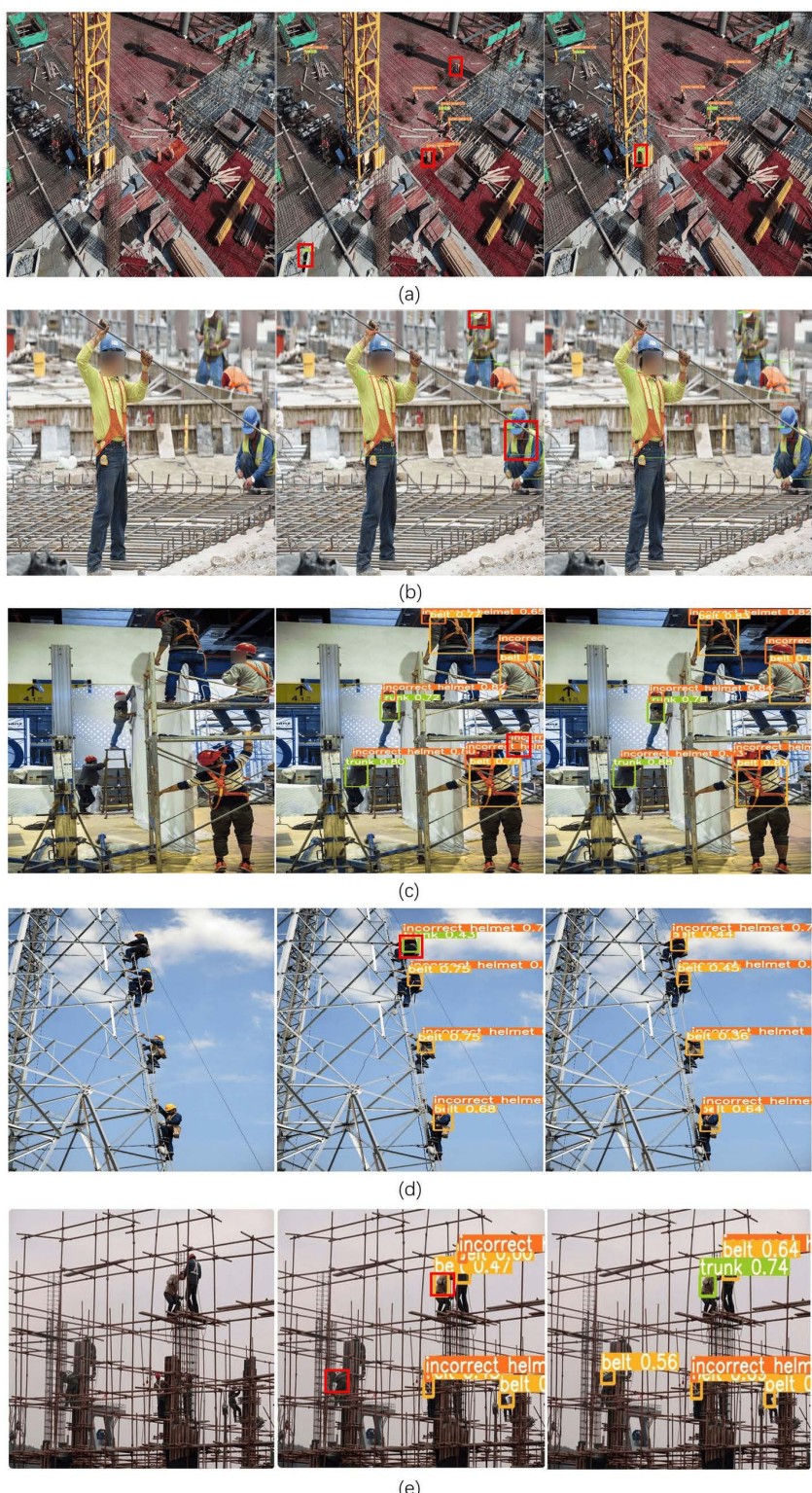

(a)

(b)

(c)

(d)

(e)

**Fig 21. Comparison of detection results.** The first column of images is the image to be detected, the second column of images is the original YOLOv5n detection effect image, the third column of images is the YOLO-CGBSE detection effect image, Fig (a) is the UAV viewpoint image, Fig (b) is the high-resolution image, Fig (c) is the indoor operation image, Fig (d) is the high altitude operation image, and Fig (e) is the occluded high altitude operation image.

before and after the improvement, which is caused by the size of the dataset being too small and the morphology of the detection target being too homogeneous. The detection effect is shown in Fig 22.

Fig 22 shows that both YOLO-M and YOLOv5n have extremely high confidence in detecting safety guardrails, but when detecting very dense safety nets, YOLOv5n encounters problems with missed detections, which is enough to prove that the improvement in this paper is very effective.

## Deploying the model and building the system

The detection system built in this study uses the embedded AI computing platform Jetson Nano B01 as the main control core. The other hardware components include a power module, a high-definition camera, and an IPS screen. The system framework diagram is shown in Fig 23.

The environment and code are debugged, all the external devices are connected, and the device is powered by a power supply with an output of 5 V = 3 A. Its effect after debugging is shown in Fig 24.

**Table 16. Object unsafe state model performance.**

| Model | Precision% | Recall% | F1% | mAP@0.5% | mAP@0.5:0.95% | parameters | GFLOPs | FPS | FPS-j | Model size KB |
|-------|-----------|---------|-----|----------|---------------|------------|--------|-----|-------|---------------|
| YOLOv5n | 99.9 | 1 | 1.98 | 99.5 | 92. | 1770682 | 4.2 | 83.929 | 11.243 | 3906 |
| YOLO-M | 99.7 | 1 | 1.98 | 99.5 | 91.5 | 1119128 | 2.8 | 68.694 | 10.121 | 2597 |

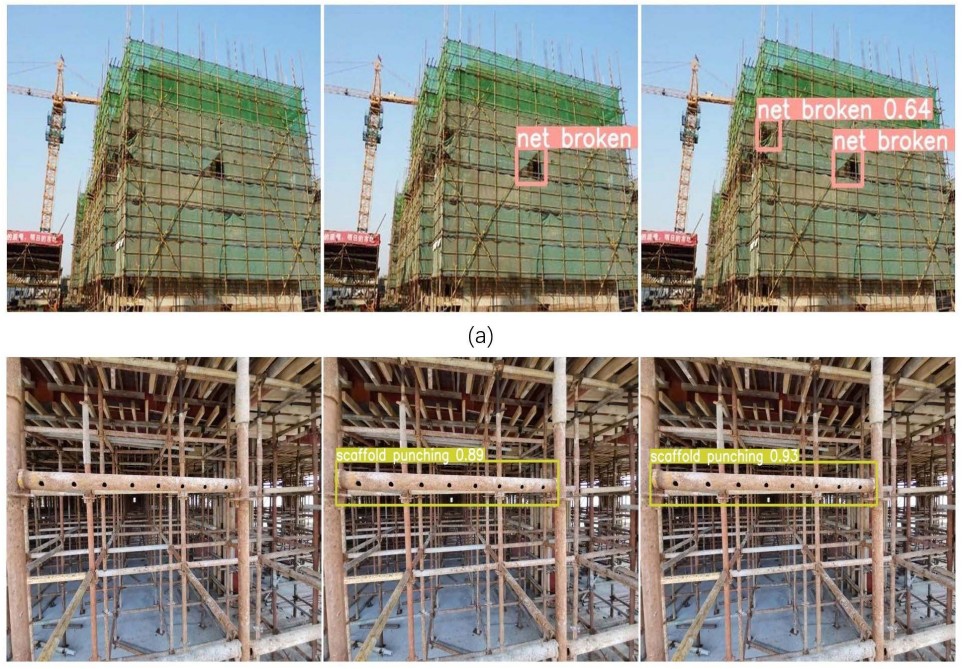

(a)

(b)

**Fig 22. Detection effect of unsafe state of object.** The first column of pictures is the picture to be detected, the second column of pictures is the original YOLOv5n detection effect, the third column is the YOLO-M detection effect, (a) is the picture of the safety net, (b) is the picture of the scaffolding perforation.

Since tasks such as image and video detection can be performed by high-performance desktop computers, this paper deploys the system onsite to detect the surrounding environment in real time through cameras and tests the real-time detection performance of the intelligent hazard identification system for construction sites. The detection results are shown in Fig 25.

The detection accuracy of this system is sufficient to meet practical needs. However, in strong light environments, the intelligent hazard identification system for construction sites has difficulty detecting small targets at a distance, which will be a future research direction.

## Discussion and future work

Although our system performs well, it still has several limitations. First, the range of detectable hazard types remains incomplete, particularly in identifying unsafe object states. Second, although we incorporated ECA, SIoU loss, and EIoU–NMS to improve detection accuracy, these enhancements are not sufficient to fully offset the side effects of excessive model lightweighting. The system's detection accuracy also tends to decline slightly under extreme environmental conditions. Furthermore, the current system has not been integrated with existing safety protocols, such as automatically triggering alarms when hazards are detected. To address these issues, our future work will focus on expanding the system's ability to detect a broader variety of hazards, such as damaged cables, and on merging the two existing datasets into a unified dataset to enhance the model's generalizability. We also aim to further improve the detection accuracy—especially under extreme conditions—by designing new modules. Additionally, we will explore ways to integrate the system with existing safety protocols while safeguarding worker privacy, enabling functionalities such as tiered alerts and evidence logging through automatic screenshots.

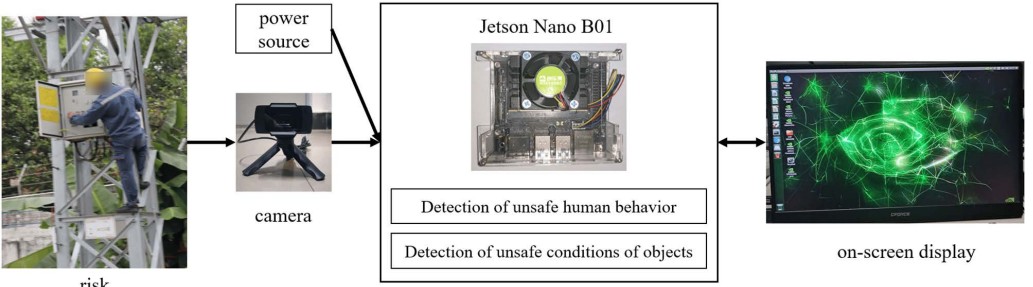

**Fig 23. System framework diagram.**

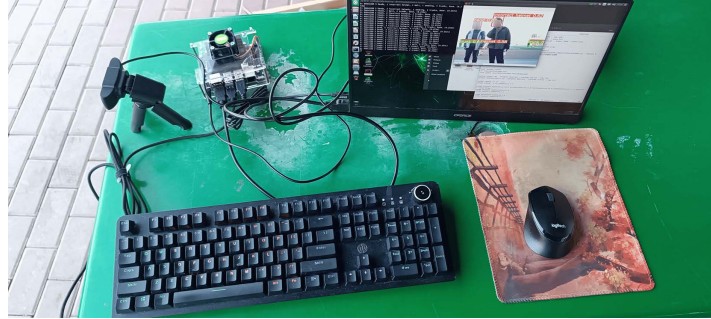

**Fig 24. Detection system display.**

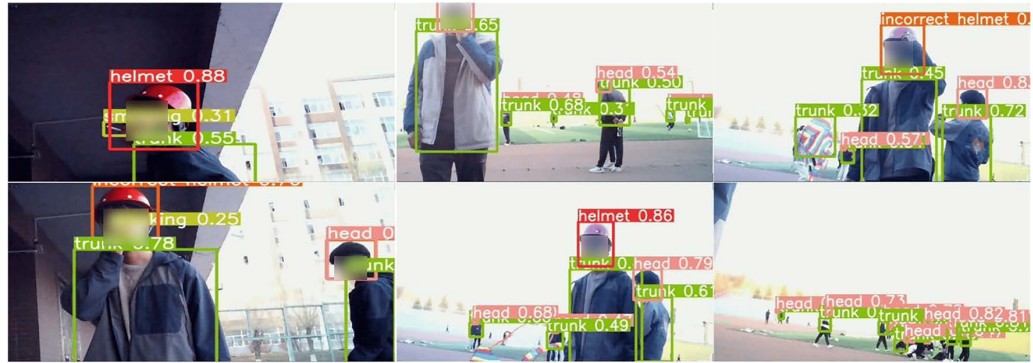

**Fig 25. Identify the system's detection results.**

## Conclusion

In this study, we propose an intelligent hazard identification system for construction sites. We first improved YOLOv5n using ECA, ghost, SIoU loss, and EIoU–NMS to increase the model's detection speed and accuracy while reducing missed detection rates. We then trained two detection models using a self-constructed dataset. Notably, the sizes of our models are 2497 KB and 2597 KB, with mAP@0.5% reaching 93.6% and 99.5%, respectively. Finally, these two detection models were deployed on the Jetson Nano B01 SUB embedded deep learning platform, creating a construction site hazard detection system capable of effectively performing hazard detection tasks. Although the dataset of unsafe behaviors involving people, which is the most comprehensive of its kind available, includes many types of unsafe behaviors, there are still some hazards not included, such as insulated gloves and shoes for live work. More importantly, datasets for unsafe states of objects have not been studied by any researchers to date, and relevant images on the internet are scarce. Therefore, we collected relevant images onsite. However, owing to the impact of the pandemic and the negative effects of such actions on construction sites, we managed to gather only a very limited number of images. In summary, our research has made a significant contribution to the field of hazard detection at construction sites. By detecting hazards in real time at construction sites, we can significantly reduce accident rates, improve the efficiency of safety audits, and provide quantifiable, traceable data support for construction site safety management. We believe that these findings will lead to more effective safety management practices and provide multifunctional applications in various fields.

## Author contributions

**Conceptualization:** Hang Li, Peijian Jin.

**Data curation:** Long Zhan, Weilong Yan.

**Formal analysis:** Peijian Jin, Long Zhan, Shimei Sun.

**Funding acquisition:** Peijian Jin, Shimei Sun.

**Investigation:** Long Zhan, Weilong Yan.

**Methodology:** Hang Li, Peijian Jin.

**Project administration:** Hang Li, Peijian Jin.

**Resources:** Hang Li, Peijian Jin.

**Software:** Hang Li, Weilong Yan, Shihao Guo.

**Supervision:** Long Zhan.

**Validation:** Hang Li, Weilong Yan, Shihao Guo.

**Visualization:** Weilong Yan, Shihao Guo.

**Writing – original draft:** Hang Li.

**Writing – review & editing:** Peijian Jin, Shimei Sun.

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
