## [Decision Letter · Decision Letter 0]

10 Mar 2025

Dear Dr. Jin,

We look forward to receiving your revised manuscript.

Kind regards,

Fatih Uysal, Ph.D.

Academic Editor

PLOS ONE

2. In your Methods section, please include additional information about your dataset and ensure that you have included a statement specifying whether the collection and analysis method complied with the terms and conditions for the source of the data.

4. We suggest you thoroughly copyedit your manuscript for language usage, spelling, and grammar. If you do not know anyone who can help you do this, you may wish to consider employing a professional scientific editing service.

“the Ministry of Emergency Management of the People's Republic of China Grant numbers:Jilin-0050-2018AQ.”

6. In the online submission form you indicate that your data is not available for proprietary reasons and have provided a contact point for accessing this data. Please note that your current contact point is a co-author on this manuscript. According to our Data Policy, the contact point must not be an author on the manuscript and must be an institutional contact, ideally not an individual. Please revise your data statement to a non-author institutional point of contact, such as a data access or ethics committee, and send this to us via return email. Please also include contact information for the third party organization, and please include the full citation of where the data can be found.

7. We note that Figures 9, 20, 20, 21 and 24 in your submission contain copyrighted images. All PLOS content is published under the Creative Commons Attribution License (CC BY 4.0), which means that the manuscript, images, and Supporting Information files will be freely available online, and any third party is permitted to access, download, copy, distribute, and use these materials in any way, even commercially, with proper attribution. For more information, see our copyright guidelines: http://journals.plos.org/plosone/s/licenses-and-copyright.

1. You may seek permission from the original copyright holder of Figures 9, 20, 20, 21 and 24 to publish the content specifically under the CC BY 4.0 license.

Additional Editor Comments:

Please revise the paper according to my comments below in addition to the referee comments.

1. YOLO v5 was preferred for object detection operations in the study. Please explain in detail why v5 was used in particular, although it is much more up-to-date in the literature (such as YOLO v12).

2. The literature review part of the study is very limited. This part definitely needs to be detailed. In addition, a literature table consisting of columns such as "pros and cons, originality, results, used dataset" should be added.

3. Like many metrics in object detection, the choice of loss function is also very important. For this reason, please explain in detail how the two different SIOU-loss and EIOU-loss used in this study were determined and whether any customization was made at this point.

4. There are very serious deficiencies in the evaluation metrics section. For this reason, metrics such as Average Recall (AR), Optimal Localization Recall Precision (oLRP), Precision-Recall Curve and Count of predicted bounding box should also be obtained.

5. In order to compare the results and reveal the superiority of the proposed approach in the literature, it is very important to obtain and compare the results for this dataset with several state-of-the-art models.

Reviewers' comments:

Reviewer's Responses to Questions

**Comments to the Author**

1. Is the manuscript technically sound, and do the data support the conclusions?

Reviewer #1: Yes

Reviewer #2: Yes

2. Has the statistical analysis been performed appropriately and rigorously?

Reviewer #1: Yes

Reviewer #2: Yes

3. Have the authors made all data underlying the findings in their manuscript fully available?

Reviewer #1: No

Reviewer #2: Yes

4. Is the manuscript presented in an intelligible fashion and written in standard English?

Reviewer #1: No

Reviewer #2: Yes

Reviewer #1: Need some improvements. Please check file. Please include all this improvement in your paper. Acknowledge the limitations of the current study, such as potential biases in the dataset or overfitting due to the model complexity.

Reviewer #2: Clarity in Problem Statement

Comment: "Ensure that the problem statement is clearly defined. While the paper mentions hidden dangers, it would be helpful to specifically define what qualifies as a 'hidden danger' on construction sites (e.g., poorly stored equipment, unsafe scaffolding, electrical hazards)."

2. Data and Dataset Clarification

Comment: "Provide more details on the dataset used. Are the images annotated manually or are they sourced from existing databases? A brief explanation of how the data was collected and pre-processed could enhance the reader's understanding."

3. YOLO Implementation

Comment: "The paper mentions using YOLO for object detection. Please clarify which version of YOLO (YOLOv3, YOLOv4, YOLOv5, etc.) was used and why this version was chosen over others, especially in the context of construction site recognition."

4. Accuracy and Performance Metrics

Comment: "The paper could benefit from more detailed performance metrics, such as precision, recall, F1 score, or mAP (mean Average Precision). A comparison of YOLO's performance versus other potential detection algorithms could provide more context for its efficacy in this application."

5. Challenges and Limitations

Comment: "While the system seems promising, it would be good to address any limitations encountered during implementation. For example, how does the model handle occlusion or cluttered environments typical of construction sites? Are there any false positive or false negative challenges?"

6. Real-Time Application

Comment: "The real-time application of this system is mentioned but not elaborated upon. Can the system be deployed effectively in real-world scenarios with live video feeds from construction sites? A brief discussion of the computational efficiency or hardware requirements would be valuable."

7. Safety Considerations

Comment: "It would be useful to briefly discuss how the system could be integrated with existing safety protocols on construction sites. How will the system alert workers or managers to detected hidden dangers? Are there considerations for worker privacy or data security?"

8. Further Improvements

Comment: "Consider adding a section on potential future work. For instance, could this system be expanded to detect additional types of hazards (e.g., environmental conditions like weather or lighting changes)? What are the next steps for improving detection accuracy?"

9. Related Work

Comment: "Please consider including a brief comparison to related work or similar AI-based hazard detection systems. How does the proposed system stand out in terms of accuracy, scalability, or real-time detection?"

10. Conclusion

Comment: "The conclusion should not only summarize the findings but also emphasize the practical implications of the system for improving construction site safety. A brief mention of its potential impact on reducing accidents or improving safety audits could strengthen the conclusion."

**Do you want your identity to be public for this peer review?** For information about this choice, including consent withdrawal, please see our Privacy Policy

Reviewer #1: No

Reviewer #2: **Yes: ** DR. SHERIN ZAFAR

---

## [Author Response · Author response to Decision Letter 1]

8 Aug 2025

Reply to journal requirements:

Concern 1: Please ensure that your manuscript meets PLOS ONE's style requirements, including those for file naming. The PLOS ONE style templates can be found at https://journals.plos.org/plosone/s/file?id=wjVg/PLOSOne_formatting_sample_main_body.pdf and https://journals.plos.org/plosone/s/file?id=ba62/PLOSOne_formatting_sample_title_authors_affiliations.pdf.

Author response: Thank you for your reminder. The authors have carefully reviewed PLOS ONE's formatting requirements and will ensure the manuscript fully complies with the journal's guidelines, including the file naming specifications. I have downloaded and consulted PLOS ONE's style template and will double-check all details before submission.

Concern 2: In your Methods section, please include additional information about your dataset and ensure that you have included a statement specifying whether the collection and analysis method complied with the terms and conditions for the source of the data.

Author response: Thank you for your feedback. The authors have supplemented additional information regarding the dataset in the "Method" section as requested, and have explicitly stated that the data collection and analysis methods fully comply with the terms and conditions of the data source. The relevant modifications are reflected in the “Dataset description” section of the “Method” section of the manuscript.

Concern 3: Please note that PLOS ONE has specific guidelines on code sharing for submissions in which author-generated code underpins the findings in the manuscript. In these cases, we expect all author-generated code to be made available without restrictions upon publication of the work. Please review our guidelines at https://journals.plos.org/plosone/s/materials-and-software-sharing#loc-sharing-code and ensure that your code is shared in a way that follows best practice and facilitates reproducibility and reuse.

Author response: Thank you for your reminder. The authors have carefully reviewed PLOS ONE's code sharing policy and fully understand the journal's requirements regarding the sharing of author-generated code. To ensure the reproducibility and reusability of the code, the authors have shared both the code and dataset on Figshare (DOI: https://doi.org/10.6084/m9.figshare.28644194.v1). Additionally, the generated code has been uploaded to GitHub at: https://github.com/1520133943/YOLO-based-intelligent-recognition-system-for-hidden-dangers-on-construction-sites.

Concern 4: We suggest you thoroughly copyedit your manuscript for language usage, spelling, and grammar. If you do not know anyone who can help you do this, you may wish to consider employing a professional scientific editing service.

Author response: Thank you for your valuable suggestions. The authors have purchased editing services from AJE to improve the English expression of the manuscript. The editing certificate has been uploaded as an attachment to the system.

Concern 5: Thank you for stating the following financial disclosure:

“the Ministry of Emergency Management of the People's Republic of China Grant numbers:Jilin-0050-2018AQ.”

Author response: Thank you for your reminder. Regarding the role of the funder, the following statement is now declared: "The Ministry of Emergency Management of the People's Republic of China (Approval No.: Jilin-0050-2018AQ) provided funding support for this study. The funder had no role in study design, data collection and analysis, decision to publish, or preparation of the manuscript." The authors have added the above statement to the cover letter and confirmed its accuracy.

Concern 6: In the online submission form you indicate that your data is not available for proprietary reasons and have provided a contact point for accessing this data. Please note that your current contact point is a co-author on this manuscript. According to our Data Policy, the contact point must not be an author on the manuscript and must be an institutional contact, ideally not an individual. Please revise your data statement to a non-author institutional point of contact, such as a data access or ethics committee, and send this to us via return email. Please also include contact information for the third party organization, and please include the full citation of where the data can be found.

Author response: Thank you for your reminder. The authors have revised the data statement according to your request. The data and code have been publicly shared on Figshare with the DOI https://doi.org/10.6084/m9.figshare.28644194.v1, and the code has been uploaded to GitHub at https://github.com/1520133943/YOLO-based-intelligent-recognition-system-for-hidden-dangers-on-construction-sites. Therefore, the data can be accessed directly via Figshare without the need for additional contacts.

Concern 7: We note that Figures 9, 20, 20, 21 and 24 in your submission contain copyrighted images. All PLOS content is published under the Creative Commons Attribution License (CC BY 4.0), which means that the manuscript, images, and Supporting Information files will be freely available online, and any third party is permitted to access, download, copy, distribute, and use these materials in any way, even commercially, with proper attribution. For more information, see our copyright guidelines: http://journals.plos.org/plosone/s/licenses-and-copyright.

1. You may seek permission from the original copyright holder of Figures 9, 20, 20, 21 and 24 to publish the content specifically under the CC BY 4.0 license.

Author response: Thank you for your valuable comments. Figures 9, 10, 20, and 21 in the manuscript are sourced from Google and public datasets such as the SHWD dataset. Many studies, especially those related to helmet detection, have used these images, which are not subject to copyright. Therefore, the authors believe that it is unnecessary to provide written permission for their use. The individuals depicted in Figure 24 are the authors themselves, so no ethical statement is required. Overall, this study involves only the authors and does not involve any other participants.

Reply to Additional Editor Comments:

Concern 1: YOLO v5 was preferred for object detection operations in the study. Please explain in detail why v5 was used in particular, although it is much more up-to-date in the literature (such as YOLO v12).

Author response: Thank you to the editor for the valuable comments. As of now, the YOLO series algorithms have been updated to the 12th version (YOLOv12), but YOLOv5 remains the most suitable for this study due to the limited computing power of the Jetson Nano B01 used to build the system. Although later versions such as YOLOv6 to YOLOv12 offer stronger performance, they require significantly more computational resources and cannot run smoothly—or even properly—on the Jetson Nano B01 (as confirmed by updated experiments). YOLOv5 has fewer model parameters and lower computational demands, making it suitable for devices with limited performance, as demonstrated by the updated experiments. More importantly, YOLOv5 offers excellent deployment compatibility, has a very active community, and benefits from numerous tutorials, pre-trained models, and third-party tool support. Since it was released earlier, many developers have built various applications and optimization tools based on YOLOv5. This study focuses more on algorithm application rather than breakthroughs in algorithm performance, so the authors chose the more stable YOLOv5. The authors also explain this in detail in the YOLOv5 section of the Methods section of the revised manuscript.

Concern 2: The literature review part of the study is very limited. This part definitely needs to be detailed. In addition, a literature table consisting of columns such as "pros and cons, originality, results, used dataset" should be added.

Author response: Thank you for the editor’s valuable comments. In the revised manuscript, the authors have rewritten the literature review section to provide a more comprehensive overview of key literature in the relevant field. In addition, following your suggestion, we have added a comparison table of the literature (Table 1), which covers aspects such as Datasets, Method, Originality, Results, and Limitations.

Concern 3: Like many metrics in object detection, the choice of loss function is also very important. For this reason, please explain in detail how the two different SIOU-loss and EIOU-loss used in this study were determined and whether any customization was made at this point.

Author response: Thank you for the editor’s valuable suggestions. Regarding the choice of loss functions, the authors mainly referred to widely used mainstream loss functions such as EIOU-loss and SIOU-loss in existing literature. Based on this research foundation, the authors conducted systematic experimental comparisons to select the loss function most suitable for this study. It should be noted that no customized modifications were made to the loss functions in this research. The authors have added relevant explanations to the Loss function section in the Method section of the manuscript.

Concern 4: There are very serious deficiencies in the evaluation metrics section. For this reason, metrics such as Average Recall (AR), Optimal Localization Recall Precision (oLRP), Precision-Recall Curve and Count of predicted bounding box should also be obtained.

Author response: Thank you for the editor’s insightful comments regarding the evaluation metrics. The authors have carefully considered your suggestions and made the following revisions in the experimental section:

The authors have supplemented the Precision (P) curve, Recall (R) curve, PR curve, and F1 curve (see Figure 22 in the revised manuscript).

Regarding the optimal localization recall precision (oLRP) and predicted bounding box count, the authors fully understand the reviewer’s suggestions. However, due to the following considerations, these two metrics were not added in the revised manuscript:

Currently, the core evaluation metrics for YOLO series algorithm research mainly include Precision, Recall, mAP, FPS, etc. Although oLRP is a relatively new comprehensive evaluation metric, its calculation process and analysis methods have not yet been widely standardized in the current research field and are not fully compatible with the evaluation framework used in this work, which may introduce additional uncertainties. The predicted bounding box count is more suitable for multi-stage optimization or scenarios requiring precise control of the number of boxes, whereas this study mainly focuses on end-to-end detection accuracy, making this metric of limited reference value in the current setting. Recent related literature in your journal’s field has also not employed these metrics, for example:

Research on mechanical automatic food packaging defect detection model based on improved YOLOv5 algorithm, https://doi.org/10.1371/journal.pone.0321971

Bearing defect detection based on the improved YOLOv5 algorithm, https://doi.org/10.1371/journal.pone.0310007.

Since these studies focus on non-construction fields, the authors did not include them in the revised manuscript; however, their underlying algorithms are the same, so they adopt the same performance metrics. The authors ensure that the evaluation metrics used are sufficient to comprehensively reflect the performance of this method. The authors hope for your understanding.

Concern 5: In order to compare the results and reveal the superiority of the proposed approach in the literature, it is very important to obtain and compare the results for this dataset with several state-of-the-art models.

Author response: Thank you for the valuable suggestion from the editor. The authors fully agree that introducing advanced models for comparison helps to more clearly demonstrate the superiority of the proposed method. Therefore, in the revised manuscript, the authors have supplemented the Model Comparison section with several current mainstream advanced models as comparison targets, including the lightweight versions of YOLOv7 to YOLOv12, and conducted systematic experimental comparisons on the same dataset.

The relevant results are shown in Table 15, which compares the performance of each method across metrics such as Precision, Recall, F1, mAP@0.5, mAP@0.5:0.95, parameters, GFLOPs, FPS, FPS on portable devices, and model size. The experimental results demonstrate that the proposed method in this study significantly outperforms existing methods in terms of compatibility and portability, validating its effectiveness and advancement.

Reply to review comments to the author:

Reply to reviewers #1:

Reviewer #1: Need some improvements. Please check file. Please include all this improvement in your paper. Acknowledge the limitations of the current study, such as potential biases in the dataset or overfitting due to the model complexity.

Author response: Thank you for your valuable comments. The authors have carefully considered your suggestions and have made corresponding revisions and improvements to the manuscript. In the Discussion and future work section of the revised version, the authors acknowledge the current limitations of the study and propose directions for future work.

Concern 1: The abstract is well-structured but can be more concise. Consider summarizing the contributions in fewer words to avoid redundancy.

Author response: Thank you to Reviewer 1 for your recognition and suggestions regarding the abstract. The authors have carefully revi

---

## [Editor Report · Decision Letter 1]

26 Aug 2025

YOLO-based intelligent recognition system for hidden dangers at construction sites

PONE-D-25-09028R1

Dear Dr. Jin,

We’re pleased to inform you that your manuscript has been judged scientifically suitable for publication and will be formally accepted for publication once it meets all outstanding technical requirements.

Kind regards,

Fatih Uysal, Ph.D.

Academic Editor

PLOS ONE

Additional Editor Comments (optional):

The article was accepted because the revisions satisfactorily addressed the referee comments, and the work shows strong potential to contribute to the literature.
---

## [Editor Report · Acceptance letter]

PONE-D-25-09028R1

PLOS ONE

Dear Dr. Jin,

I'm pleased to inform you that your manuscript has been deemed suitable for publication in PLOS ONE. Congratulations! Your manuscript is now being handed over to our production team.

Kind regards,

on behalf of

Dr. Fatih Uysal

Academic Editor

PLOS ONE